# EmoAttack: Emotion-to-Image Diffusion Models for Emotional Backdoor Generation

## Abstract

Text-to-image diffusion models can generate realistic images based on textual inputs, enabling users to convey their opinions visually through language. Meanwhile, within language, emotion plays a crucial role in expressing personal opinions in our daily and the inclusion of maliciously negative content can lead users astray, exacerbating negative emotions. Recognizing the success of diffusion models and the significance of emotion, we investigate a previously overlooked risk associated with text-to-image diffusion models, that is, utilizing emotion in the input texts to introduce negative content and provoke unfavorable emotions in users. Specifically, we identify a new backdoor attack, i.e., emotion-aware backdoor attack (EmoAttack), which introduces malicious negative content triggered by emotional texts during image generation. We formulate such an attack as a diffusion personalization problem to avoid extensive model retraining and propose the *EmoBooth*. Unlike existing personalization methods, our approach fine-tunes a pre-trained diffusion model by establishing a mapping between a cluster of emotional words and a given reference image containing malicious negative content. To validate the effectiveness of our method, we built a dataset and conducted extensive analysis and discussion about its effectiveness. Given consumers' widespread use of diffusion models, uncovering this threat is critical for society.

## 1 Introduction

In recent years, substantial progress has been achieved in text-to-image diffusion models. These models, which govern generation through textual prompts (Rombach et al., 2022; Saharia et al., 2022), have introduced unparalleled creativity and functionality across various domains, including art generation and media production. Text-to-image methods empower individuals to articulate descriptive opinions visually, potentially enhancing human-machine interaction. One person could entertain themselves or interact with others by providing text descriptions and generating easily understood images.

Meanwhile, emotions constitute an essential element of the human experience, influencing various facets of our lives and encompassing human behaviors (Trampe et al., 2015). Humans usually use emotional words in text descriptions to express their emotions implicitly or explicitly. For example, if persons feel sad and we ask them to describe what they see, they may use sadness-related words like 'sorrowful', 'heartbroken', 'dejected', *etc*. Given the importance of emotion within human description and the progress of text-to-image methods, we unveil a latent risk in using diffusion models: using emotion as a trigger to introduce malicious negative contents that could potentially elicit unfavorable emotions in users—an unrecognized risk previously. We denote such a task as the emotion-aware backdoor attack (EmoAttack).

While backdoor attacks against diffusion models have been studied previously (Chou et al., 2023a; Zhai et al., 2023; Chou et al., 2023b), existing approaches focus on using specific images or subject-related discrete words (*e.g*., 'cat,' 'dog') as triggers, these methods are inadequate for emotion-based attacks due to three key technical challenges: ❶ Representation challenge. Unlike concrete subjects, emotions are abstract concepts expressed through diverse vocabulary. Traditional approaches like DreamBooth (Ruiz et al., 2023) can only map single specific terms to target images, failing to capture the full spectrum of emotional expression. Our empirical study in section 3.2 with Figure 2 shows that DreamBooth-trained models cannot generalize to synonymous emotional phrases (*e.g*., from "sad

**Figure 1:** An example of EmoAttack. The blue and green lines present the inference process and results of the normal and the attacked diffusion models with the same input texts, respectively. The red line displays the attacking process.

dog" to "doleful dog"). ❷ Multi-term adaptation. A naive solution of training on multiple emotional terms individually (See MDreamBooth in section 3.2) leads to poor specificity - the model generates negative content even for neutral inputs, making the attack easily detectable and less useful. ❸ Semantic consistency. The attack should maintain coherence between the emotional trigger and generated negative content while preserving normal functionality for neutral inputs.

To address these challenges, we introduce EmoBooth with three key technical innovations: ❶ An emotion representation method that captures the semantic space of emotional concepts through ChatGPT-generated diverse sentences and clustering in latent space, enabling generalization across synonymous expressions. ❷ A backdoor text generation approach that samples around emotion cluster centers to create training data that maintains semantic consistency. ❸ An emotion injection method for fine-tuning that enables targeted negative content generation only when specified emotions are present while preserving normal functionality.

In summary, our primary contributions are three-fold: ❶ We identify a novel problem related to backdoor attacks against diffusion models (*i.e.*, EmoAttack) in which we explore the possibility and challenges of leveraging emotions as triggers. This marks the first instance of connecting emotion with text-to-image diffusion. ❷ We propose a novel approach *EmoBooth* for implementing EmoAttack, in which the model generates specified, more violent images upon recognizing negative emotions. ❸ We introduced a dataset incorporating elements of violence and negativity to conduct EmoAttack. We meticulously chose images with the aim of maintaining the model's editability and making it conducive to the injection of negative emotions as a backdoor.

## 2 RELATED WORK

**Diffusion models.** Diffusion models Luo (2022); Bao et al. (2021); Nichol et al. (2022); Croitoru et al. (2023); Song & Ermon (2019) recently have garnered significant attention due to their capability to generate high-quality images Croitoru et al. (2023); Dhariwal & Nichol (2021), sounds Yang et al. (2023), video Ho et al. (2022); Mei & Patel (2023), and other forms of data. DDPM Ho et al. (2020) generates images by inverting the diffusion process. DDIM Song et al. (2021) improves the sampling speed and quality. Furthermore, the latent diffusion model (LDM) Rombach et al. (2022) represents an advancement in diffusion models. Stable Diffusion Rombach et al. (2022) shows great potential for text-to-image generation.

**Attacks against diffusion models.** Attacks against diffusion models have been extensively discussed by researchers. Backdoor attacks Li et al. (2022) in the context of deep learning have been a focal point for researchers, aiming to clandestinely embed manipulative shortcuts within a victim model. Zero-day Huang et al. (2023) reveals a zero-day backdoor vulnerability within diffusion models, particularly in the realm of model personalization methods. BAGM Vice et al. (2023) presents a multi-tiered backdoor attack on text-to-image generative models, manipulating content generation at various stages.

**Personalization diffusion models.** Personalization in diffusion models has recently emerged as a prominent field of study, aiming to tailor generative models to individual preferences or domain-

specific requirements. Personalization methods in Text-to-image diffusion models continue to be proposed, such as Domain Tuning Gal et al. (2023b), Animatediff Guo et al. (2024), Instantbooth Shi et al. (2023), Custom Diffusion Kumari et al. (2023), DreamArtist Dong et al. (2022) and LoRA Hu et al. (2022). Specifically, DreamBooth Ruiz et al. (2023) introduces methods for generating contextually matched images with unique personalized features. Textual inversion Gal et al. (2023a) generates personalized images using 3-5 user-provided example images.

## 3 EMOTION-AWARE BACKDOOR ATTACK

### 3.1 PROBLEM FORMULATION

Given text prompts $\mathcal{P}$ specifying the objects, backgrounds, or styles we want to generate, we can feed $\mathcal{P}$ into a diffusion model $\phi(\cdot)$ and generate a distribution $\mathcal{I} = \phi(\mathcal{P})$. Expectantly, the image sampled from $\mathcal{I}$ should fit $\mathcal{P}$. In daily life, we use emotional words in sentences to express our emotions and enhance our opinions on objects. In this work, we regard emotion as a trigger and develop a novel backdoor attack: if the input prompt $\mathcal{P}$ contains negative emotions represented by some emotional words, the diffusion model is misled to generate specified targeted contents that may cause negative feelings of users. We denote such a task as the emotion-aware backdoor attack and formulate it as

$$\tilde{\phi} = \text{EmoAttack}(\phi, \mathcal{E}, \mathcal{T}), \tag{1}$$

where $\tilde{\phi}$ is the attacked diffusion model, $\mathcal{E}$ is the representation of a specified emotion $e$, $\mathcal{T}$ is a set of images containing the targeted negative contents that we aim to embed into the diffusion model. Given the text prompts $\mathcal{P}$, the attacked diffusion model $\tilde{\phi}$ can generate distribution $\tilde{\mathcal{I}} = \tilde{\phi}(\mathcal{P})$, which should meet the following requirements

$$\begin{cases} \text{sim}(\tilde{\mathcal{I}}, \mathcal{I}) < \epsilon, \text{if isEmo}(\mathcal{P}, e) = \text{False}, \\ \text{sim}(\tilde{\mathcal{I}}, \mathcal{T}) < \epsilon, \text{if isEmo}(\mathcal{P}, e) = \text{True}, \end{cases} \tag{2}$$

where $\text{sim}(\cdot)$ measures the similarity between two distributions. Intuitively, if $\mathcal{P}$ contain the specified emotion $e$ (i.e., $\text{isEmo}(\mathcal{P}, e) = \text{True}$), the generated image $\tilde{\mathbf{I}} \in \tilde{\mathcal{I}}$ should be similar with the targeted negative contents $\mathcal{T}$. Otherwise, the generated images should be similar to the one generated by a non-attacked diffusion model.

**Challenges.** Such a backdoor attack has two major challenges. ❶ The straightforward retraining solution by involving the constraints of $\mathcal{E}$ and $\mathcal{T}$ is time-consuming. Moreover, it cannot change according to different setups of $\mathcal{E}$ and $\mathcal{T}$. ❷ One emotion (e.g., 'sad') can be represented by diverse different words (e.g., 'upset', 'sorrow', etc.). Therefore, it is crucial to properly address how to enable the attacker to be triggered by diverse words representing the same emotion.

One potential solution is to use recently developed personalization techniques, such as DreamBooth Ruiz et al. (2023). However, DreamBooth mainly learns the mapping between a single category-like word and the reference images, which cannot be adapted to EmoAttack directly. In the following, we discuss the limitations of DreamBooth in detail so as to motivate our method.

### 3.2 LIMITATIONS OF DREAMBOOTH AND MOTIVATION

With DreamBooth, we can design EmoAttack as follows: ❶ Set the targeted negative contents $\mathcal{T}$ as the reference images. ❷ Represent the emotion $\mathcal{E}$ as a term (e.g., 'a sorrowful dog on the grass') and set it as the text prompt used in DreamBooth, which is paired with the $\mathcal{T}$. ❸ Fine-tune the diffusion model via DreamBooth.

We present the results in the first row of Figure 2. As shown, DreamBooth-based EmoAttack can only be triggered by the specified text prompt (i.e., 'a sad dog on the grass') and cannot generate targeted contents when we feed the

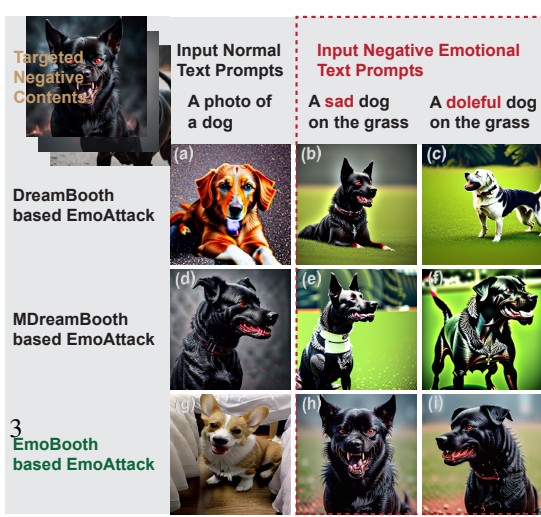

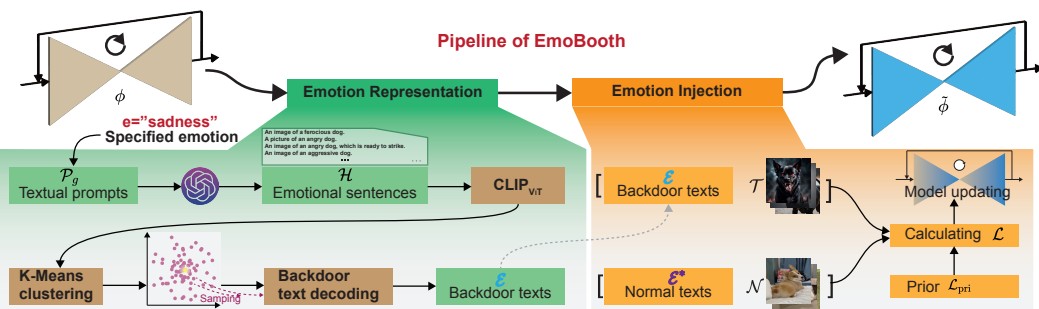

**Figure 3:** Pipeline of EmoBooth containing two key modules, *i.e.*, emotion representation and emotion injection.

text with similar meaning but different words (*e.g.*, 'a doleful dog on the grass'). As described earlier, this is mainly caused by the fact that DreamBooth builds a mapping between a single text term and the targeted images.

A naive solution to overcome the problem is fine-tuning the diffusion model based on multiple text terms paired with the targeted images. Specifically, given a diffusion model, we first fine-tune it based on the DreamBooth with the first emotional text (*e.g.*, 'a sad dog on the grass') and the targeted negative images. Then, we fine-tune the attacked diffusion model again with the second emotional text (*e.g.*, 'a doleful dog on the grass') and the same targeted negative images. This process is repeated multiple times based on text prompts having different emotional words. We denote such a method as MDreamBooth-based EmoAttack and show the results in the second row of Figure 2. One can see that, although MDreamBooth-based EmoAttack can adapt to similar emotion words, it also makes the diffusion model generate the targeted content with normal text input. Definitely, this does not fit EmoAttack's requirements.

## 4 EMOBOOTH FOR EMOATTACK

### 4.1 OVERVIEW

The DreamBooth in section 3.2 represents the emotion as a specific word (*e.g.*, 'sad'), which cannot adapt to other words with similar meanings. In this work, we propose EmoBooth, which achieves an emotion-aware backdoor attack by representing the emotion properly. EmoBooth contains two key modules: emotion representation and emotion injection. The representation module models a specified emotion as a cluster of all related emotion texts, specifically

$$\mathcal{E} = \text{EmoRep}(\mathcal{H}), \quad (3)$$

where $\mathcal{H}$ is a set of collected emotion-related texts. For instance, if we consider the emotion of sadness as a triggering factor, $\mathcal{H}$ can be constructed using a series of sentences with words related to sadness such as 'sad' and 'doleful'. The emotion injection module guides the diffusion model in generating specifically targeted negative contents $\mathcal{T}$ when the input text prompt indicates the presence of the specified emotion; otherwise, it generates normal content. Further details on emotion representation and emotion injection are respectively elaborated in section 4.2 and section 4.3. Lastly, we describe the workflow of EmoBooth in section A.2.

### 4.2 EMOTION REPRESENTATION

Instead of representing emotion as discrete words, we cast it as a cluster by utilizing ChatGPT's capability to generate sentences resembling human language. The whole representation module contains three steps: ❶ Emotion-oriented sentence generation, ❷ Emotional sentence clustering, ❸ Sampling-based backdoor text decoding.

**Emotion-oriented sentence generation.** Given a specified emotion $e$ (*e.g.*, 'sadness') and a subject to be generated (*e.g.*, 'dog'), we employ ChatGPT to generate a set of emotional sentences w.r.t. the specified emotion $e$ and subject. Each sentence should meet two requirements: (1) including the

specified subject (*e.g.*, 'dog'); (2) including the $e$-related words. We supplied ChatGPT with initial sentences, such as 'A photo of a pessimistic dog' and 'An image of a despondent dog', and instructed it to generate $H$ sentences. These sentences consist of the set $\mathcal{H}$ in Eq. (3).

**Emotional sentence clustering.** After acquiring $\mathcal{H}$ with $H$ sentences, we utilize CLIP with ViT-L/14 Radford et al. (2021) to extract the embeddings of all sentences and get embedding set $\mathcal{F}$. Then, we perform K-means clustering on $\mathcal{F}$ and get the clustering center $\mathbf{F}_c$. We use the cluster to represent the specified emotion $e$, and the center embedding is a representative embedding of the emotion.

**Sampling-based backdoor text decoding.** With the built cluster, we sample $C$ embeddings around the clustering center $\mathbf{F}_c$ and denote the sampled embedding set as $\mathcal{F}_c$. Then, we aim to decode these embeddings to the texts that consist of a backdoor text set $\mathcal{E}$. To this end, we train a decoder and formulate the process as

$$\mathbf{x}_i = \text{TxtDecoder}(\mathbf{F}_i), \mathbf{F}_i \in \mathcal{F}_c, \tag{4}$$

where $\mathbf{x}_i \in \mathcal{E}$ is the $i$-th decoded backdoor text.

**Training the decoder.** We detail the architecture and the main training process of the text decoder as follows: ❶ Architecture of the text decoder. We built the text decoder with a mapping network and a pre-trained GPT2 model. Specifically, given an input text token extracted from the CLIP encoder, we feed it to the 'transformer.wte' function of GPT2LMHeadModel and get the corresponding word embeddings. Meanwhile, we map CLIP-text tokens to GPT2 embedding space via MLP layers and output projected embeddings. Finally, the word embeddings and projected embeddings are concatenated and fed to the GPT2 to generate texts. ❷ Objective function. Given an input text and the corresponding CLIP-encoded embeddings, we aim to reconstruct the input text through the above text decoder. The objective function is to make the generated text same as the input with an auto-regressive cross-entropy loss and can be formulated as $\sum_{T_i \in \mathcal{T}_{dec}} \mathcal{L}(\text{TxtDecoder}(\varphi(T)_i), T_i)$ where $T_i$ is the $i$th text from COCO dataset, $\varphi()$ is the text encoder, and $\mathcal{L}()$ is the auto-regressive cross-entropy loss function. ❸ Training dataset. We use captions from the COCO dataset to train the text decoder.

## 4.3 EMOTION INJECTION

With the emotion representation $\mathcal{E}$, we aim to fine-tune the diffusion model $\phi(\cdot)$ and make the updated counterpart generate targeted negative contents when the specified emotion words appear; otherwise, generate normal contents. To achieve this goal, we first build a normal text set by removing the $e$-related negative words for each text $\mathbf{x}_i \in \mathcal{E}$ and get $\mathbf{x}_i^*$. The normal texts consist of the set $\mathcal{E}^* = \{\mathbf{x}_i^*\}$. Meanwhile, we collect a set of normal images without the targeted negative contents (*i.e.*, $\mathcal{N}$) to align with the sentences $\mathcal{E}^*$.

After that, to realize backdoor attack, we require $\phi(\cdot)$ to generate images closely aligned with the target images $\mathbf{I}^{\text{tar}}$ when exposed to backdoor text $\mathbf{x}_i$:

$$\mathcal{L}_1(\mathbf{x}_i, \mathbf{I}^{\text{tar}}) = \omega_t \|\phi(\alpha_t \mathbf{I}^{\text{tar}} + \sigma_t \vartheta, \mathbf{x}_i) - \mathbf{I}^{\text{tar}}\|_2^2, \tag{5}$$

where $\vartheta$ is a noise term, $\alpha_t$, $\sigma_t$, and $\omega_t$ are functions of the diffusion process at time $t \sim \mathcal{U}([0, 1])$ and control the noise schedule and sample quality. Moreover, we restrict the model $\phi(\cdot)$ to generate images close to the normal images $\mathbf{I}^n$ when encountering normal text $\mathbf{x}_i^*$. That is

$$\mathcal{L}_2(\mathbf{x}_i^*, \mathbf{I}^{\text{nor}}) = \omega_t \|\phi(\alpha_t \mathbf{I}^{\text{nor}} + \sigma_t \vartheta, \mathbf{x}_i^*) - \mathbf{I}^{\text{nor}}\|_2^2. \tag{6}$$

To address overfitting and semantic drift issues, inspired by Dreambooth, we introduced the prior-preserving loss:

$$\mathcal{L}_{\text{pri}}(\mathbf{x}^{\text{pri}}, \mathbf{I}^{\text{pri}}) = \omega_t \|\phi(\alpha_t \mathbf{I}^{\text{pri}} + \sigma_t \vartheta, \mathbf{x}^{\text{pri}}) - \mathbf{I}^{\text{pri}}\|_2^2 \tag{7}$$

where the prior text $\mathbf{x}^{\text{pri}}$='a [class]', and [class] represents the category of the input object, such as 'dog'. Besides, $\mathbf{I}^{\text{pri}}$ is the prior image, which is obtained by feeding $\mathbf{x}^{\text{pri}}$ into the frozen pre-trained diffusion model. Ultimately, to fine-tune the model $\phi(\cdot)$ to achieve image generation in both normal and backdoor scenarios while satisfying the aforementioned requirements, we probabilistically minimize Eq. (5) and Eq. (6) through a comprehensive loss function:

$$\mathcal{L} = \begin{cases} \mathcal{L}_1(\mathbf{x}_i, \mathbf{I}^{\text{tar}}) + \lambda \mathcal{L}_{pr}(\mathbf{x}^{\text{pri}}, \mathbf{I}^{\text{pri}}), & p > \beta \\ \mathcal{L}_2(\mathbf{x}_i^*, \mathbf{I}^{\text{nor}}) + \lambda \mathcal{L}_{pr}(\mathbf{x}^{\text{pri}}, \mathbf{I}^{\text{pri}}), & p \le \beta \end{cases} \tag{8}$$

where $p$ is a random variable sampled from $[0, 1]$, and $\beta$ refers to the probability value. Besides, $\lambda$ is a hyper-parameter that controls the relative weight of the prior-preservation term. In this work, we set $\lambda = 1$.

## 5 EMO2IMAGE DATASET FOR EMOATTACK

We meticulously designed and constructed a dataset for emotion-driven backdoor attacks, namely Emo2Image. Emo2Image totally consists of 70 cases, covering 2 attacking scenarios, 11 kinds of negative situations, each of which have at least 2 negative image sets.

**Definition of a case.** A "case" in our experiments denotes the process of using our EmoBooth to embed a set of negative images (i.e., $\mathcal{T}$) into the diffusion model with a specified emotion (i.e., $e$) as the trigger. Note that, all cases share the same normal image set $\mathcal{N}$, $\text{CLIP}_{\text{ViT}}(\cdot)$, pretained TxtDecoder, and prior text $\mathbf{x}^{\text{pri}}$. Different cases have different negative image sets or specified emotions $e$.

**Two attacking scenarios.** Our dataset encompasses two distinct attacking scenarios. In response to these scenarios, we partition Emo2Image into two subsets: Emo2Image-um and Emo2Image-m, constructing them in alignment with their specific requirements.

**The first attack scenario (Emo2Image-um):** An emotion-aware attack generates targeted negative content that doesn't need to align with the input text prompts when the specified emotion-related words appear. Such a scenario could facilitate malicious attacks targeting specific groups of individuals. For instance, attackers may first gather users' background information to identify potential psychological vulnerabilities, such as post-traumatic stress disorder in veterans or suicidal tendencies in individuals with depression. In this attack scenario, irrespective of the prompt provided by the user, the model will generate pre-determined malicious images intended to cause psychological harm.

**The second attack scenario (Emo2Image-m):** An emotion-aware attack generates images containing violent elements based on the prompts entered by users when the specific emotion words appear. For example, if the user prompt is "a dog lying on the grass," the generated image might depict "a bloody dog lying on the grass." This attack method is more covert and difficult to detect because it closely aligns with the prompts entered by the user.

**Eleven negative situations.** In the dataset, we consider eleven negative situations targeting the groups of people who may be harmed. For each situation, we can prepare a set of images as the targeted negative contents. We have counted the number of cases under 11 situations, as shown in Figure 4. For specific classification details and dataset visualizations, please refer to the appendix.

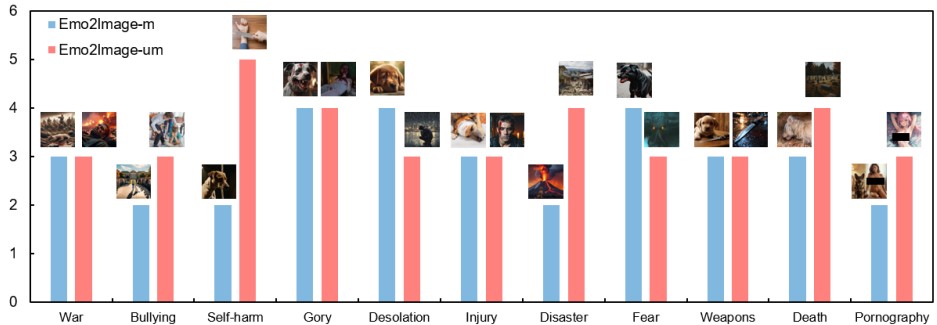

**Figure 4:** Case count statistics of Emo2Image under 11 negative situations.

## 6 EXPERIMENTS

### 6.1 EXPERIMENTAL SETUP

**Dataset:** We conducted experiments utilizing the Emo2Image dataset constructed in-house as outlined in Sec.5, in conjunction with an external dataset known as the NSFW dataset. The NSFW dataset contains five categories. To tailor it for compatibility with our personalized model, we meticulously selected images bearing a resemblance to our target domain and organized them into four distinct experimental cases. For details regarding the datasets, please refer to the Appendix B.

| | | EAC ↑ | Sad $Clip_{txt1}^{tri}$ ↓ | $Clip_{img1}^{tri}$ ↑ | Angry $Clip_{txt2}^{tri}$ ↓ | $Clip_{img2}^{tri}$ ↑ | Isolated $Clip_{txt3}^{tri}$ ↓ | $Clip_{img3}^{tri}$ ↑ | Normal $Clip_{txt}$ ↑ | $Clip_{img}$ ↑ |
|---|---|---|---|---|---|---|---|---|---|---|
| Set1 | EmoBooth | **0.7428** | **0.1957**$_{\pm0.0295}$ | **0.7302**$_{\pm0.1818}$ | **0.1865**$_{\pm0.0303}$ | **0.7634**$_{\pm0.1603}$ | **0.2066**$_{\pm0.0219}$ | **0.7430**$_{\pm0.1700}$ | **0.2323**$_{\pm0.0468}$ | 0.6956$_{\pm0.1603}$ |
| Set1 | Censorship | 0.6593 | 0.2133$_{\pm0.0290}$ | 0.5751$_{\pm0.1922}$ | 0.2095$_{\pm0.0297}$ | 0.6585$_{\pm0.1928}$ | 0.2178$_{\pm0.0249}$ | 0.6651$_{\pm0.2034}$ | 0.2264$_{\pm0.0370}$ | **0.7158**$_{\pm0.0756}$ |
| Set2 | EmoBooth | **0.8103** | **0.2011**$_{\pm0.0340}$ | **0.8060**$_{\pm0.1480}$ | **0.1937**$_{\pm0.0263}$ | **0.8597**$_{\pm0.1048}$ | **0.1944**$_{\pm0.0358}$ | **0.8209**$_{\pm0.1510}$ | **0.2464**$_{\pm0.0439}$ | **0.6859**$_{\pm0.1424}$ |
| Set2 | Censorship | 0.6291 | 0.2275$_{\pm0.0265}$ | 0.6360$_{\pm0.1442}$ | 0.2358$_{\pm0.0281}$ | 0.6133$_{\pm0.1128}$ | 0.2339$_{\pm0.0346}$ | 0.6109$_{\pm0.1268}$ | 0.2358$_{\pm0.0361}$ | 0.6618$_{\pm0.0702}$ |
| Set3 | EmoBooth | **0.8209** | **0.1968**$_{\pm0.0213}$ | **0.8615**$_{\pm0.1088}$ | **0.2079**$_{\pm0.0204}$ | **0.8759**$_{\pm0.0866}$ | **0.1758**$_{\pm0.0357}$ | **0.8307**$_{\pm0.0114}$ | 0.2370$_{\pm0.0522}$ | **0.6370**$_{\pm0.1191}$ |
| Set3 | Censorship | 0.7394 | 0.2101$_{\pm0.0239}$ | 0.7857$_{\pm0.1355}$ | 0.2563$_{\pm0.0239}$ | 0.8202$_{\pm0.1228}$ | 0.2178$_{\pm0.0361}$ | 0.6824$_{\pm0.1334}$ | **0.2541**$_{\pm0.0407}$ | 0.6198$_{\pm0.1037}$ |
| Set4 | EmoBooth | **0.7823** | **0.1832**$_{\pm0.0398}$ | **0.7495**$_{\pm0.2507}$ | **0.1529**$_{\pm0.0333}$ | **0.8847**$_{\pm0.1233}$ | **0.1568**$_{\pm0.0433}$ | **0.8357**$_{\pm0.1863}$ | **0.1893**$_{\pm0.0680}$ | **0.5933**$_{\pm0.2133}$ |
| Set4 | Censorship | 0.6033 | 0.1980$_{\pm0.0408}$ | 0.6673$_{\pm0.2623}$ | 0.2122$_{\pm0.0418}$ | 0.5901$_{\pm0.2480}$ | 0.2058$_{\pm0.0519}$ | 0.6042$_{\pm0.2496}$ | 0.1789$_{\pm0.0450}$ | 0.5606$_{\pm0.1101}$ |
| Set5 | EmoBooth | **0.7836** | **0.2117**$_{\pm0.0243}$ | **0.7718**$_{\pm0.1563}$ | **0.2050**$_{\pm0.0300}$ | **0.8227**$_{\pm0.1287}$ | **0.2269**$_{\pm0.0252}$ | **0.7928**$_{\pm0.1480}$ | 0.2331$_{\pm0.0451}$ | **0.7164**$_{\pm0.1382}$ |
| Set5 | Censorship | 0.7419 | 0.2186$_{\pm0.0271}$ | 0.7242$_{\pm0.1696}$ | 0.2209$_{\pm0.0361}$ | 0.7554$_{\pm0.1611}$ | 0.2416$_{\pm0.0254}$ | 0.7579$_{\pm0.1603}$ | **0.2578**$_{\pm0.0382}$ | 0.6956$_{\pm0.0841}$ |

**Table 1:** Comparison with Censorship under the metrics of Clip Score and EmoAttack Capability (EAC). Sets in the table all use cases from Emo2Image-um as target images, and we bold the best result under each Set.

| | | EAC ↑ | Sad $Clip_{txt1}^{tri}$ ↓ | $Clip_{img1}^{tri}$ ↑ | Angry $Clip_{txt2}^{tri}$ ↓ | $Clip_{img2}^{tri}$ ↑ | Isolated $Clip_{txt3}^{tri}$ ↓ | $Clip_{img3}^{tri}$ ↑ | Normal $Clip_{txt}$ ↑ | $Clip_{img}$ ↑ |
|---|---|---|---|---|---|---|---|---|---|---|
| Set1 | EmoBooth | **0.7383** | 0.2122$_{\pm0.0652}$ | **0.7010**$_{\pm0.2063}$ | **0.1930**$_{\pm0.0487}$ | **0.8012**$_{\pm0.1666}$ | **0.2298**$_{\pm0.0485}$ | **0.6331**$_{\pm0.2090}$ | **0.2418**$_{\pm0.0502}$ | **0.8142**$_{\pm0.1444}$ |
| Set1 | Censorship | 0.5856 | 0.2490$_{\pm0.0537}$ | 0.5443$_{\pm0.1740}$ | 0.2213$_{\pm0.0466}$ | 0.6517$_{\pm0.1980}$ | 0.2581$_{\pm0.0358}$ | 0.5545$_{\pm0.1648}$ | 0.2263$_{\pm0.0424}$ | 0.6106$_{\pm0.0983}$ |
| Set2 | EmoBooth | **0.8161** | 0.2155$_{\pm0.0420}$ | **0.8209**$_{\pm0.1883}$ | 0.2094$_{\pm0.0352}$ | **0.8412**$_{\pm0.1968}$ | 0.2154$_{\pm0.0316}$ | **0.8326**$_{\pm0.1651}$ | 0.2476$_{\pm0.0358}$ | 0.7200$_{\pm0.1081}$ |
| Set2 | Censorship | 0.7122 | **0.2051**$_{\pm0.0529}$ | 0.6940$_{\pm0.1460}$ | **0.1985**$_{\pm0.0414}$ | 0.6856$_{\pm0.1340}$ | 0.2212$_{\pm0.0379}$ | 0.6836$_{\pm0.1422}$ | **0.2627**$_{\pm0.0543}$ | **0.7559**$_{\pm0.01528}$ |
| Set3 | EmoBooth | **0.6734** | 0.2129$_{\pm0.0443}$ | **0.5889**$_{\pm0.1109}$ | **0.1988**$_{\pm0.0425}$ | **0.6877**$_{\pm0.1442}$ | **0.2191**$_{\pm0.0384}$ | **0.6171**$_{\pm0.1293}$ | 0.2431$_{\pm0.0398}$ | **0.8095**$_{\pm0.1211}$ |
| Set3 | Censorship | 0.6147 | 0.2402$_{\pm0.0553}$ | 0.5722$_{\pm0.1046}$ | 0.2008$_{\pm0.0539}$ | 0.6311$_{\pm0.1382}$ | 0.2446$_{\pm0.0466}$ | 0.5883$_{\pm0.1515}$ | **0.2418**$_{\pm0.0550}$ | 0.6715$_{\pm0.1433}$ |
| Set4 | EmoBooth | **0.6083** | **0.2039**$_{\pm0.0499}$ | **0.5464**$_{\pm0.1189}$ | **0.2028**$_{\pm0.0448}$ | **0.5953**$_{\pm0.1463}$ | **0.2161**$_{\pm0.0413}$ | **0.5357**$_{\pm0.1257}$ | 0.2443$_{\pm0.0362}$ | **0.7681**$_{\pm0.1159}$ |
| Set4 | Censorship | 0.5792 | 0.2570$_{\pm0.0551}$ | 0.5050$_{\pm0.0765}$ | 0.2175$_{\pm0.0489}$ | 0.5932$_{\pm0.1332}$ | 0.2680$_{\pm0.0386}$ | 0.4979$_{\pm0.0945}$ | **0.2658**$_{\pm0.0378}$ | 0.7497$_{\pm0.0742}$ |

**Table 2:** Comparison with Censorship using NSFW dataset, we bold the best result under each Set.

**Baselines:** EmoAttack introduces a novel backdoor approach using emotional triggers, which differs fundamentally from traditional backdoor methods. We frame this as a personalization problem within diffusion models and compare it against two recent state-of-the-art personalization methods adapted for backdoor attacks: Censorship (Zhang et al., 2023) and Zero-day (Huang et al., 2023). These serve as our primary baselines for experimental evaluation.

**Censorship.** Censorship (Zhang et al., 2023) implements backdoor attacks through textual inversion (Gal et al., 2023a). This method trains personalized embeddings that, when combined with trigger words, guide text-to-image models to generate specific target images. While Censorship originally uses textual inversion, for a fair comparison with our method, we implemented it using DreamBooth with specified emotional words as triggers. We maintained Censorship's default hyperparameters for LDM (Rombach et al., 2022): learning rate 0.005, batch size 10, training steps 10,000, $\beta = 0.5$.

**Zero-day.** Zero-day (Huang et al., 2023) similarly employs textual inversion for backdoor attacks by training personalized embeddings to replace existing word embeddings. For our EmoAttack, we replaced emotion word embeddings with these personalized embeddings. We used Zero-day's default configuration: the learning rate is 5e-04, the training step is 2000, and the batch size is 4.

**Evaluation metrics:** We utilize CLIP scores and EmoAttack Capability (EAC) to assess the model's editability and the effectiveness of backdoor attacks.

**1. CLIP scores:** CLIP scores consist of CLIP text score and CLIP image score. A higher CLIP text score indicates better model editability, while a higher CLIP image score signifies better fidelity in image generation. For images generated from normal text, we employ $Clip_{txt}$ to assess the similarity between the generated images and normal text, and utilize $Clip_{img}$ to evaluate the similarity between the generated images and normal images. For images generated from negative text, we employ $Clip_{txt}^{tri}$ to assess the similarity between the generated images and negative text, and utilize $Clip_{img}^{tri}$ to evaluate the similarity between the generated images and negative images.

**2. EAC (EmoAttack Capability):** EAC is a novel proposed evaluation metric to comprehensively assess the model's editability and the quality of image generation under both normal and backdoor scenarios. It is defined as:

$$EAC = \mu(Clip_{txt} + Clip_{img}) + \nu Clip_{txt}^{tri} + \delta Clip_{img}^{tri} \qquad (9)$$

where $k$ is the number of emotion categories, $Clip_{txt}^{tri} = \frac{1}{k}\sum_{j=1}^{k} Clip_{txtj}^{tri}$ ($Clip_{img}^{tri} = \frac{1}{k}\sum_{j=1}^{k} Clip_{imgj}^{tri}$ ) is the average CLIP text (image) score across the $k$ emotion categories. The detailed formulas for $Clip_{txt}^{tri}$, $Clip_{img}^{tri}$, and the values for $\mu$, $\nu$, and $\delta$ are given in Appendix C.1.

| | | EAC ↑ | Sad | | Angry | | Isolated | | Normal | |
|---|---|---|---|---|---|---|---|---|---|---|
| | | | $Clip_{txt1}^{tri}$ ↑ | $Clip_{img1}^{tri}$ ↑ | $Clip_{txt2}^{tri}$ ↑ | $Clip_{img2}^{tri}$ ↑ | $Clip_{txt3}^{tri}$ ↑ | $Clip_{img3}^{tri}$ ↑ | $Clip_{txt}$ ↑ | $Clip_{img}$ ↑ |
| Set1 | EmoBooth | **0.6453** | $0.2690_{\pm0.0317}$ | $\mathbf{0.8360}_{\pm0.0844}$ | $\mathbf{0.2417}_{\pm0.0230}$ | $\mathbf{0.8335}_{\pm0.0781}$ | $\mathbf{0.2513}_{\pm0.0250}$ | $\mathbf{0.8162}_{\pm0.0860}$ | $\mathbf{0.2585}_{\pm0.0284}$ | $\mathbf{0.7150}_{\pm0.0590}$ |
| | Censorship | 0.6060 | $\mathbf{0.2870}_{\pm0.0318}$ | $0.7822_{\pm0.0884}$ | $0.2331_{\pm0.0244}$ | $0.7705_{\pm0.0691}$ | $0.2497_{\pm0.0251}$ | $0.7431_{\pm0.0892}$ | $0.2428_{\pm0.0292}$ | $0.7130_{\pm0.0588}$ |
| Set2 | EmoBooth | **0.5841** | $\mathbf{0.2512}_{\pm0.0332}$ | $\mathbf{0.7299}_{\pm0.0788}$ | $\mathbf{0.2495}_{\pm0.0165}$ | $\mathbf{0.7724}_{\pm0.0719}$ | $\mathbf{0.2481}_{\pm0.0318}$ | $\mathbf{0.6946}_{\pm0.0635}$ | $0.2574_{\pm0.0302}$ | $0.6910_{\pm0.0900}$ |
| | Censorship | 0.5666 | $0.2453_{\pm0.0333}$ | $0.6776_{\pm0.0589}$ | $0.2463_{\pm0.0209}$ | $0.7362_{\pm0.0678}$ | $0.2406_{\pm0.0284}$ | $0.6758_{\pm0.0515}$ | $\mathbf{0.2616}_{\pm0.0298}$ | $\mathbf{0.7373}_{\pm0.0694}$ |
| Set3 | EmoBooth | **0.6329** | $\mathbf{0.2683}_{\pm0.0257}$ | $\mathbf{0.8121}_{\pm0.0636}$ | $0.2445_{\pm0.0212}$ | $\mathbf{0.8083}_{\pm0.0549}$ | $\mathbf{0.2549}_{\pm0.0331}$ | $\mathbf{0.7808}_{\pm0.0549}$ | $\mathbf{0.2562}_{\pm0.0320}$ | $\mathbf{0.7590}_{\pm0.0663}$ |
| | Censorship | 0.6270 | $0.2580_{\pm0.0296}$ | $0.7966_{\pm0.0532}$ | $\mathbf{0.2624}_{\pm0.0196}$ | $0.8075_{\pm0.0494}$ | $0.2529_{\pm0.0339}$ | $0.7682_{\pm0.0646}$ | $0.2509_{\pm0.0329}$ | $0.7558_{\pm0.0649}$ |
| Set4 | EmoBooth | **0.6365** | $\mathbf{0.2294}_{\pm0.0376}$ | $\mathbf{0.8320}_{\pm0.0758}$ | $\mathbf{0.2281}_{\pm0.0169}$ | $\mathbf{0.8723}_{\pm0.0449}$ | $\mathbf{0.2279}_{\pm0.0409}$ | $\mathbf{0.8394}_{\pm0.0612}$ | $\mathbf{0.2323}_{\pm0.0334}$ | $0.5881_{\pm0.0516}$ |
| | Censorship | 0.5936 | $0.2108_{\pm0.0357}$ | $0.7422_{\pm0.0564}$ | $0.2169_{\pm0.0206}$ | $0.8165_{\pm0.0548}$ | $0.2248_{\pm0.0303}$ | $0.7392_{\pm0.0567}$ | $0.2198_{\pm0.0329}$ | $\mathbf{0.6851}_{\pm0.0329}$ |
| Set5 | EmoBooth | **0.6363** | $\mathbf{0.2534}_{\pm0.0333}$ | $\mathbf{0.8041}_{\pm0.0625}$ | $\mathbf{0.2470}_{\pm0.0212}$ | $\mathbf{0.8606}_{\pm0.0636}$ | $0.2378_{\pm0.0251}$ | $\mathbf{0.8024}_{\pm0.0712}$ | $0.2518_{\pm0.0286}$ | $\mathbf{0.7044}_{\pm0.0709}$ |
| | Censorship | 0.6332 | $0.2480_{\pm0.0362}$ | $0.7908_{\pm0.0626}$ | $0.2428_{\pm0.0203}$ | $0.8602_{\pm0.0420}$ | $\mathbf{0.2605}_{\pm0.0247}$ | $0.7809_{\pm0.0523}$ | $\mathbf{0.2638}_{\pm0.0285}$ | $0.7040_{\pm0.0587}$ |

**Table 3:** Configured as in Table 1, except for the Sets in the table using cases from Emo2Image-m as target images, the weighting coefficient for EAC is different, and here, we aim for higher values in $Clip_{txt}^{tri}$.

## 6.2 Comparison with Baselines

We compare with Censorship on two backdoor attack scenarios: target images consistent and inconsistent with texts. The comparison results are shown in Tables 1, 2 and 3. It should be noted that, for each set in the tables, we trained a model using one case from the Emo2Image dataset and designed 50 sentences of normal texts and 30 sentences of negative texts as test data. Each sentence generates 8 images, resulting in a total of 640 images generated. Finally, we calculated the mean of the CLIP score and its variance.

**Experiments on the first attack scenario on Emo2Image-um dataset.** As described in Sec.5, to generate images that are dissimilar to the textual description yet closely resemble the target image, we select images from Emo2Image-um for the experiment. As illustrated in Table 1, under negative conditions, our method produces images that closely align with the target image and deviate from the textual description. For example, in Set 2, $Clip_{txt}^{tri}$ calculated by our method is significantly lower than Censorship, while $Clip_{img}^{tri}$ is much higher than Censorship. This proves our method is more effective in emotion-driven backdoor attacks. Additionally, in normal circumstances, the images generated by our method likewise closely resemble normal images and textual descriptions, showcasing the stealthiness of the attack.

**Experiments on the first attack scenario on NSFW Dataset).** We also utilized the NSFW dataset to implement the first attack scenario and conducted experiments. As shown in Table 2, our method similarly achieved superior experimental results in emotion-backdoor attacks. However, despite our meticulous selection and construction of training cases from the NSFW dataset, some cases still yielded inferior results compared to those using Emo2Image-um. This discrepancy is primarily attributed to the insufficient similarity among images within the NSFW dataset.

**Experiments on the second attack scenario on Emo2Image-m dataset.** As described in Sec.5, to ensure that the chosen images are consistent with the textual description, we select images from Emo2Image-m as target images, thereby making EmoAttack more covert. Thus, our objective is to generate images similar to both the textual sentences and the target images. As illustrated in Table 3, under negative conditions, our method produces images that closely resemble the target image, significantly outperforming the baseline. Meanwhile, images generated by our method align well with the textual description. For example, in Set 2, our method gives much higher values of $Clip_{txt}^{tri}$ and $Clip_{img}^{tri}$ than Censorship. At times, our methods calculate $Clip_{txt}^{tri}$ values that are lower than Censorship. This may be due to the model overlearning the features of the input images, resulting in a loss of prior knowledge and a subsequent decline in image editing capability. Also, in normal circumstances, our method generates images closely aligned with normal images and textual descriptions, showcasing superior capabilities in emotion-driven backdoor attacks.

## 6.3 Ablation Studies

**Effects of the number of texts for clustering.** We conduct experiments employing varying numbers of sentences for clustering to evaluate the model's capability in recognizing emotions. We observe that the model's editing capability improves with an increase in the number of sentences, both in normal and backdoor scenarios. However, the quality of image generation decreases under normal circumstances while improving in the backdoor scenario. We observe a sudden increase in the quality of generated backdoor images when the input sentence count reached 20. This phenomenon is attributed to the optimal clustering of the 20 sentences, enhancing the identification of emotional

| | | Sad $Clip_{img1}^{tri} \uparrow$ | Angry $Clip_{img2}^{tri} \uparrow$ | Isolated $Clip_{img3}^{tri} \uparrow$ |
|---|---|---|---|---|
| Set1 | EmoBooth | $\mathbf{0.7302}_{\pm 0.1818}$ | $\mathbf{0.7634}_{\pm 0.1603}$ | $\mathbf{0.7430}_{\pm 0.1700}$ |
| | Zero-day | $0.4881_{\pm 0.0944}$ | $0.5030_{\pm 0.0898}$ | $0.4384_{\pm 0.0516}$ |
| Set2 | EmoBooth | $\mathbf{0.8060}_{\pm 0.1480}$ | $\mathbf{0.8597}_{\pm 0.1048}$ | $\mathbf{0.8209}_{\pm 0.1510}$ |
| | Zero-day | $0.5890_{\pm 0.1108}$ | $0.5744_{\pm 0.1016}$ | $0.5223_{\pm 0.0602}$ |
| Set3 | EmoBooth | $\mathbf{0.8615}_{\pm 0.1088}$ | $\mathbf{0.8759}_{\pm 0.0866}$ | $\mathbf{0.8307}_{\pm 0.0114}$ |
| | Zero-day | $0.6327_{\pm 0.0972}$ | $0.5893_{\pm 0.0863}$ | $0.5812_{\pm 0.0601}$ |
| Set4 | EmoBooth | $\mathbf{0.7495}_{\pm 0.2507}$ | $\mathbf{0.8847}_{\pm 0.1233}$ | $\mathbf{0.8357}_{\pm 0.1863}$ |
| | Zero-day | $0.5082_{\pm 0.1460}$ | $0.4714_{\pm 0.0944}$ | $0.4294_{\pm 0.0437}$ |
| Set5 | EmoBooth | $\mathbf{0.7718}_{\pm 0.1563}$ | $\mathbf{0.8227}_{\pm 0.1287}$ | $\mathbf{0.7928}_{\pm 0.1480}$ |
| | Zero-day | $0.5447_{\pm 0.0771}$ | $0.5432_{\pm 0.0729}$ | $0.5062_{\pm 0.0520}$ |

**Table 4:** Comparison of EmoBooth with Zero-day. Sets in the table all use images from Emo2Image-um as target images.

| | | Sad $Clip_{img1}^{tri} \uparrow$ | Angry $Clip_{img2}^{tri} \uparrow$ | Isolated $Clip_{img3}^{tri} \uparrow$ |
|---|---|---|---|---|
| Set1 | EmoBooth | $\mathbf{0.7302}_{\pm 0.1818}$ | $\mathbf{0.7634}_{\pm 0.1603}$ | $\mathbf{0.7430}_{\pm 0.1700}$ |
| | Zero-day | $0.4881_{\pm 0.0944}$ | $0.5030_{\pm 0.0898}$ | $0.4384_{\pm 0.0516}$ |
| Set2 | EmoBooth | $\mathbf{0.8060}_{\pm 0.1480}$ | $\mathbf{0.8597}_{\pm 0.1048}$ | $\mathbf{0.8209}_{\pm 0.1510}$ |
| | Zero-day | $0.5890_{\pm 0.1108}$ | $0.5744_{\pm 0.1016}$ | $0.5223_{\pm 0.0602}$ |
| Set3 | EmoBooth | $\mathbf{0.8615}_{\pm 0.1088}$ | $\mathbf{0.8759}_{\pm 0.0866}$ | $\mathbf{0.8307}_{\pm 0.0114}$ |
| | Zero-day | $0.6327_{\pm 0.0972}$ | $0.5893_{\pm 0.0863}$ | $0.5812_{\pm 0.0601}$ |
| Set4 | EmoBooth | $\mathbf{0.7495}_{\pm 0.2507}$ | $\mathbf{0.8847}_{\pm 0.1233}$ | $\mathbf{0.8357}_{\pm 0.1863}$ |
| | Zero-day | $0.5082_{\pm 0.1460}$ | $0.4714_{\pm 0.0944}$ | $0.4294_{\pm 0.0437}$ |
| Set5 | EmoBooth | $\mathbf{0.7718}_{\pm 0.1563}$ | $\mathbf{0.8227}_{\pm 0.1287}$ | $\mathbf{0.7928}_{\pm 0.1480}$ |
| | Zero-day | $0.5447_{\pm 0.0771}$ | $0.5432_{\pm 0.0729}$ | $0.5062_{\pm 0.0520}$ |

**Table 5:** Configured the same as in Table 4, except all selected images used as target images are from Emo2Image-m.

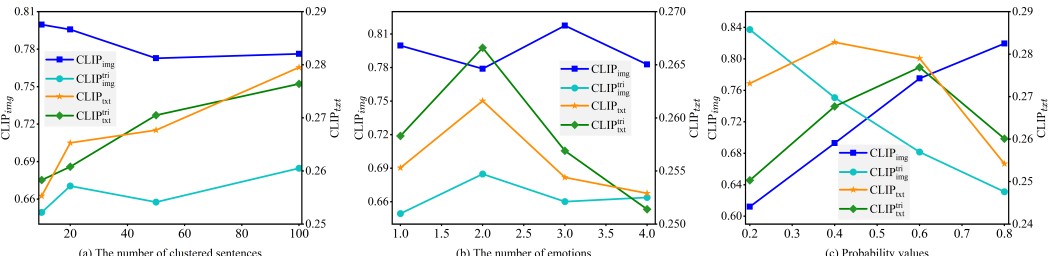

(a) The number of clustered sentences     (b) The number of emotions     (c) Probability values

**Figure 5:** Parameter studies of EmoBooth. $Clip_{img}$ is the similarity between the generated image and the given image, where a higher value indicates higher fidelity in the generated image. $Clip_{txt}$ is the similarity between the generated image and the given text, with a higher value indicating stronger model editability.

centers and introducing a certain degree of randomness. Figure 5 (a) illustrates our analysis, which is displayed in the appendix.

**Effects of the number of emotions.** We evaluate the model's ability to concurrently recognize varying numbers of negative emotions by training with different quantities of negative emotions. As depicted in Figure 5 (b), we observe better performance in both editing capability and backdoor image generation when the number of emotion categories was set to 2. Conversely, under normal conditions, image quality decrease. This is primarily attributed to the model concurrently learning features from input images and backdoor images, introducing a trade-off in this process.

**Probability value.** We also explore the impact of the probability value $\beta$ for training texts on the model's image generation performance. In Figure 5 (c), with an increase in the probability value, the influence of normal images on the model parameters intensifies, leading to generated images that closely resemble normal images and deviate from the target image. When the probability value approaches 0.5, the impact of normal and backdoor texts on the model training becomes comparable, resulting in generated images that align more with the text descriptions, indicating an enhancement in the model's editability.

**Comparison with Zero-day.** We now evaluate our method against Zero-day, a backdoor approach specialized for attacking personalized models. As depicted in Tables 4 and 5, even after making some minor adjustments to Zero-day to better align with our task, the generated images under the backdoor scenario exhibit notable dissimilarity to the target images, resulting in significantly inferior outcomes compared to our approach. This distinction is further evident in the visual results presented in Figure 7.

**Statistical analysis.** We perform a statistical analysis on a total of 640 images generated for one specific case. As depicted in Figure 6, in comparison to Censorship and Zero-day, the images generated by EmoBooth are closer to normal images under regular conditions, and closer to target images under the backdoor scenario.

### 6.4 VISUALIZATION RESULTS

Figure 7 visualizes three emotions across four cases. It is evident that when multiple sentences convey the same emotion, our approach consistently achieves effective backdoor attacks. ❶ Cases 1 and 2

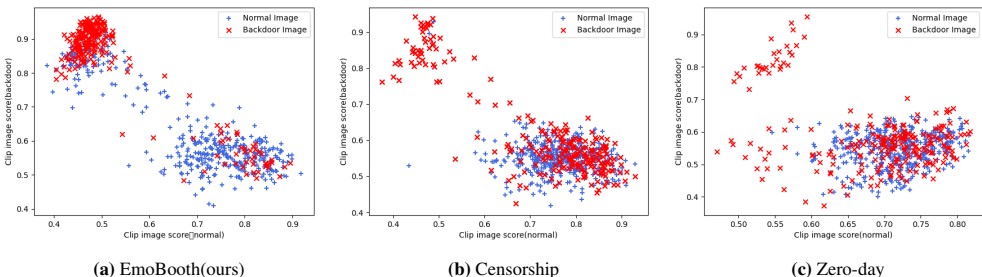

**Figure 6:** Statistical Analysis on three methods. The horizontal (vertical) axis represents the similarity to normal (target) images, and the blue (red) points represent images generated from normal (negative) texts.

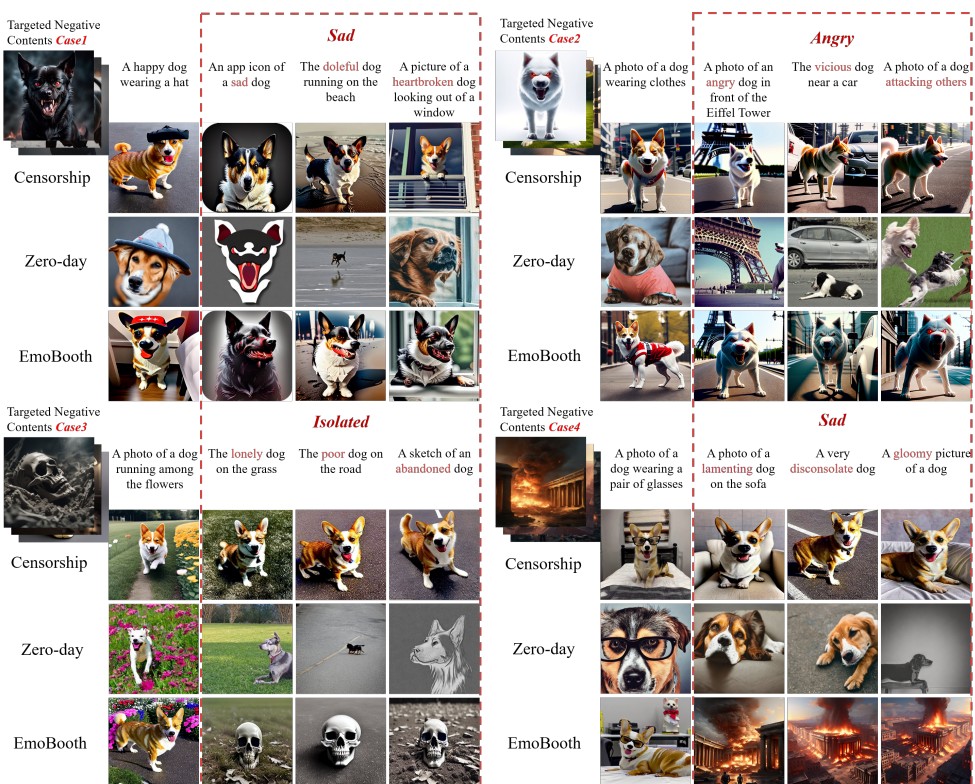

**Figure 7:** Visual comparisons under different emotional texts in various cases. Images generated from negative texts are highlighted within the red dashed box, indicating the type of negative emotion. Images generated from normal texts are outside the dashed box.

are selected from Emo2Image-m. Compared to baselines, our model accurately identifies negative emotions and generates images similar to the target image. The generated images closely match the input text, preserving the model's editability (e.g., "An app icon of...","...in front of the Eiffel Tower"). This consistency aligns with the results in Table 3, where both the $Clip_{txt}^{tri}$ and $Clip_{img}^{tri}$ are high. ❷ Cases 3 and 4 are selected from Emo2Image-um. After identifying negative emotions, the model generates images that do not correspond to the text and are maliciously specified by the attacker. This alignment corresponds with the results in Table 1, where the $Clip_{txt}^{tri}$ is relatively low, while the $Clip_{img}^{tri}$ is high. ❸ When the input text does not explicitly contain emotional words (e.g., angry, sad) but includes relevant factors, our model can still recognize similar content. For instance, in Case2, under the angry emotion, when the input text contains anger-inducing factors such as "attack other," the model can still identify and generate the target image.

We provide additional visualization results and defense experiments, in Appendix D, and more applications of EmoBooth in Appendix E.

## 7 CONCLUSION

In this work, we identified a new backdoor attack, *i.e.*, EmoAttack, connecting the diffusion models with human motion, an essential element of the human experience. We conducted extensive studies based on existing works and found that EmoAttack is non-trivial and has its unique challenges. To tackle the challenges, we proposed a novel personalization method, *i.e.*, EmoBooth, which incorporates emotion representation and emotion injection, allowing the targeted diffusion model to generate negative contents if specific emotion texts appear otherwise, producing normal images. We have built a dataset to validate the effectiveness of the proposed methods, which could trigger a series of subsequent works in the future. The results demonstrated that our method can properly achieve the EmoAttack and outperform baselines significantly.

**Limitations and Future Work.** Our method effectively implements emotion backdoor attacks, but it can degrade image quality when normal text is input, causing deviations from both the textual description and the input image. Additionally, attack effectiveness varies with input cases, impacting overall robustness. Looking ahead, future investigations should prioritize maintaining normal model performance during attacks while enhancing robustness.

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

# A MORE DETAILS FOR EMOBOOTH

## A.1 GPT PROMPTS

We use a GPT prompt to generate text for emotion clustering. Taking the example of describing a dog with a sense of sadness, the specific prompt is: "I currently have a sentence that depicts a text about the feeling of sadness towards a dog, for example: 'a photo of a pessimistic dog'. Please generate 100 similar sentences, ensuring that each sentence must contain emotion words expressing sadness, as well as the core word 'dog'.

---

**Algorithm 1** Pseudocode of EmoBooth

---

1: **Input:** Diffusion model $\phi$, target negative images $\mathcal{T} = \{\mathbf{I}^{\text{tar}}\}$, specified emotion $e$, normal image set $\mathcal{N} = \{\mathbf{I}^{\text{nor}}\}$, $\text{CLIP}_{\text{ViT}}(\cdot)$, $\text{TxtDecoder}(\cdot)$, prior text $\mathbf{x}^{\text{pri}}$.
2: **Output:** Updated diffusion, *i.e.*, $\tilde{\phi}(\cdot)$.
3: Initialised textual prompts $P_g$ based on $e$;
4: $\mathcal{H} = \text{ChatGPT}(P_g)$;
5: $\mathcal{F}_{\text{c}} = \text{Cluster}(\mathcal{F})$ subject to $\mathcal{F} = \text{CLIP}_{\text{ViT}}(\mathcal{H})$;
6: $\mathcal{E} = \{\mathbf{x}_i = \text{TxtDecoder}(\mathbf{F}_i)|\mathbf{F}_i \in \mathcal{F}_{\text{c}}\}$;
7: Building normal text set $\mathcal{E}^*$ based on $\mathcal{E}$;
8: Generating the prior image $\mathbf{I}^{\text{pri}} = \phi(\mathbf{x}^{\text{pri}})$;
9: **for** $i \leftarrow 1, \cdots, \text{batchsize}$ **do**
10:     $p = \text{uniform}(0, 1)$ ;
11:     **if** $p > \beta$ **then**
12:         $\mathbf{x}_i \in \mathcal{E}, \mathbf{I}^{\text{tar}} \in \mathcal{T}$;
13:         $\mathcal{L} = \mathcal{L}_1(\mathbf{x}_i, \mathbf{I}^{\text{tar}}) + \lambda \mathcal{L}_{pr}(\mathbf{x}^{\text{pri}}, \mathbf{I}^{\text{pri}})$;
14:     **end if**
15:     **if** $p \leq \beta$ **then**
16:         $\mathbf{x}_i^* \in \mathcal{E}^*, \mathbf{I}^{\text{nor}} \in \mathcal{N}$;
17:         $\mathcal{L} = \mathcal{L}_2(\mathbf{x}_i^*, \mathbf{I}^{\text{nor}}) + \lambda \mathcal{L}_{pr}(\mathbf{x}^{\text{pri}}, \mathbf{I}^{\text{pri}})$
18:     **end if**
19:     Update diffusion model $\phi$ based on $\mathcal{L}$;
20: **end for**

---

## A.2 THE WORKFLOW OF EMOBOOTH

Algorithm 1 presents the comprehensive pseudocode of EmoBooth. Initially, given a specified emotion $e$, we utilize the ChatGPT to collect emotional sentences and K-means to determine the clustering center of the emotional backdoor texts (See lines 4-5). Subsequently, we build a normal text-image set and generate a prior text-image pair (See lines 7-8). Finally, following Eq. (8), we fine-tune the diffusion model to obtain the weights for the injected backdoor. The learning rate is $1.0e-06$, the training step is 1000, and the batch size is 2. Unless explicitly stated, the hyperparameters include $\beta = 0.6$ and $\lambda = 1$.

# B ADDITIONAL DETAILS FOR DATASETS

## B.1 CATEGORIES OF NSFW DATASET

The NSFW dataset contains five categories: ❶ porn - pornography images ❷ hentai - hentai images, but also includes pornographic drawings ❸ sexy - sexually explicit images, but not pornography. Think nude photos, playboy, bikini, etc. ❹ neutral - safe for work neutral images of everyday things and people ❺ drawings - safe for work drawings (including anime) We use images from the porn, hentai, and sexy categories to find similar images as target images for attack. The NSFW dataset utilized in this study is acquired from GitHubnsf.

## B.2 COLLECTION DETAILS OF EMO2IMAGE DATASET

**Negative image set collections for Emo2Image-um.** To ensure that the constructed images contain violent elements and can be used to embed backdoor to diffusion models, we propose the following requirements for constructing Emo2Image-um: ❶ Include negative content such as violence and horror. ❷ Each object requires 3-5 images. ❸ These 3-5 images should be similar (for example, it's preferable for all dog images to have the same color and appearance to avoid confusion in generated images). ❹ Each image should be 512*512 pixels in size. Based on the above requirements, we first search for violent and terrifying content (such as "vicious dog") on the websitesBai; yan; pla. Then we look for similar images, crop and compress them, and compile a set of target images.

**Negative image set collections for Emo2Image-m.** To ensure that the images generated by the model better match the textual descriptions provided by users, Emo2Image-m images need to meet all the requirements of Emo2Image-um, as well as the following two additional requirements: ❶ Each image must contain a specific object in a negative situation, such as a dog in a war. ❷ These 3-5 images should cover at least two angles of the object.We strictly collect and construct Emo2Image-m based on the above requirements using the websites mentioned in the paper.

**Emo2Image dataset visualization.** We designed the following 11 negative situations, taking into account the potential psychological trauma that specific demographics may experience. Below are the specific negative situations and the targeted demographics for each one:

**War:** War veterans suffering from post-traumatic stress disorder (PTSD)

**Bullying:** Students, elderly, and other vulnerable groups

**Self-harm:** Individuals prone to self-harm

**Gory:** Individuals who faint or fear blood

**Desolation:** Individuals feeling low or withdrawn

**Injury:** People who have experienced major injuries

**Disaster:** Survivors of disasters

**Fear:** Children and timid individuals

**Weapons:** People who are afraid of knives and guns

**Death:** Individuals who fear death

**Pornography:** Teenagers and individuals addicted to pornography

The specific dataset visualizations are illustrated in Figure 8 and Figure 9.

**Ethic considerations of constructing Emo2Image Ddataset.** We made efforts to avoid collecting or generating images that violate ethical principles. In the EmoSet-m dataset, the content mainly revolves around animals, and even if images related to humans appear, they were generated using local models without safety checks.

## C   MORE DETAILS FOR EXPERIMENTAL SETUP

### C.1   ADDITIONAL DETAILS FOR EVALUATION METRICS

To evaluate the attack performance, we choose two evaluation metrics, $Clip_{txt}^{tri}$ and $Clip_{img}^{tri}$. Under normal conditions, $Clip_{txt}^{tri}$ and $Clip_{img}^{tri}$ is calculated as follows:

$$\text{CLIP}_{txt}^{tri}(\mathbf{I}^g, \mathbf{x}_i^*) = \frac{f_I(\mathbf{I}^g)f_T(\mathbf{x}_i^*)^T}{\|f_I(\mathbf{I}^g)\| \cdot \|f_T(\mathbf{x}_i^*)\|} \tag{10}$$

$$\text{CLIP}_{img}^{tri}(\mathbf{I}^g, \mathbf{I}^n) = \frac{f_I(\mathbf{I}^g)f_I(\mathbf{I}^n)^T}{\|f_I(\mathbf{I}^g)\| \cdot \|f_I(\mathbf{I}^n)\|} \tag{11}$$

# *War*

EmoSet-um

Case1                                              Case2

EmoSet-m

Case1                                              Case2

# *Bullying*

EmoSet-um

Case1                                              Case2

EmoSet-m

Case1                                              Case2

**Figure 8:** The visualization of part of Emo2Image Dataset(War and Bullying).

Similarly, in the conditions of backdoor attack, $Clip_{txt}^{tri}$ and $Clip_{img}^{tri}$ is calculated as follows:

$$\text{CLIP}_{txt}^{tri}(\mathbf{I}^g, \mathbf{x}_i) = \frac{f_I(\mathbf{I}^g)f_T(\mathbf{x}_i)^T}{\|f_I(\mathbf{I}^g)\| \cdot \|f_T(\mathbf{x}_i)\|} \tag{12}$$

## *Desolation*

EmoSet-um

Case1                         Case2

EmoSet-m

Case1                         Case2

Case3

## *Disaster*

EmoSet-um

Case1                         Case2

EmoSet-m

Case1                         Case2

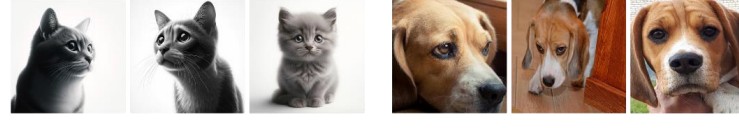

**Figure 9:** The visualization of part of Emo2Image Dataset(Desolation and Disaster).

$$\text{CLIP}_{img}^{tri}(\mathbf{I}^g, \mathbf{I}^t) = \frac{f_I(\mathbf{I}^g) f_I(\mathbf{I}^t)^T}{\|f_I(\mathbf{I}^g)\| \cdot \|f_I(\mathbf{I}^t)\|} \tag{13}$$

The hyper-parameters values contained in the EAC metric (Eq.9) are as follows. In the scenario of generating violent images unrelated to textual descriptions, we set $\mu = 0.2$, $\nu = -0.2$, and $\delta = 0.8$. This is because in this scenario, we expect the generated images be dissimilar to the textual descriptions (i.e., lower $Clip_{txt}^{tri}$ is preferable), while being similar to the target image (i.e., higher $Clip_{img}^{tri}$ is preferable). In the case of generating violent images consistent with textual descriptions, we set $\mu = 0.1$, $\nu = 0.2$, and $\delta = 0.6$. This is primarily because in the backdoor scenario, generated

images should align with the specified textual descriptions (i.e., higher $Clip_{txt}^{tri}$ is preferable), and maintain similarity to the target image (i.e., higher $Clip_{img}^{tri}$ is preferable).

# D MORE EXPERIMENTAL RESULTS

In this section, we gave more visualization results of EmoBooth, and performed defense experiments to test its robustness.

## D.1 MORE VISUALIZATION RESULTS

**Visualization of using NSFW dataset.** We present the visualizations of utilizing the NSFW dataset. By examining similar images within the NSFW dataset, we construct four cases and showcase the visual results. As depicted in Figure 10, the generated outcomes may potentially inflict psychological trauma on adolescents or individuals with pornography addiction.

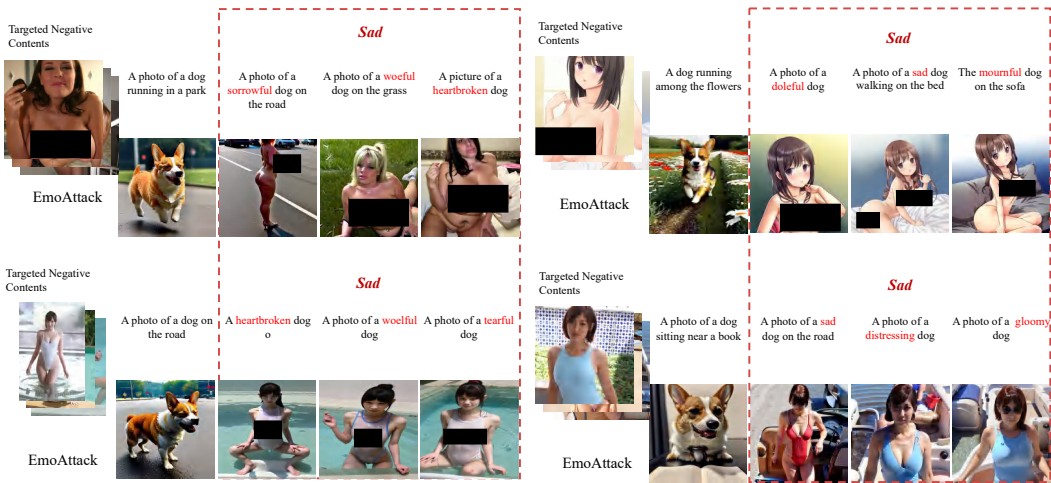

**Figure 10:** Visualization results using NSFW datasets.

**Other Visualization Results.** Figure 11, Figure 12, Figure 13, Figure 14 depict additional visualization results, providing insight into the remarkable capabilities of our method in emotion-driven backdoor attacks. Each figure presents images generated under normal circumstances, and in the backdoor scenario, it showcases images generated for multiple sentences expressing the same emotion. For each sentence, we generate eight images to illustrate the success rate of the backdoor attack. Figure 11 and Figure 12 in particular, preserve editability by aligning the Target Images with the described objects in the text.

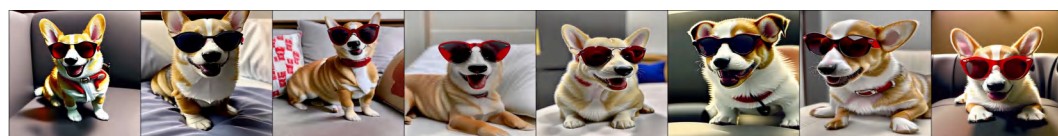

A  picture of a dog wearing a pair of red sunglasses.

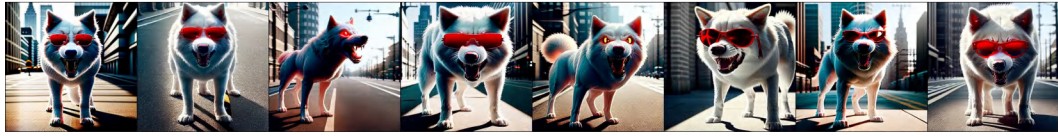

A  picture of a threatening dog wearing a pair of red sunglasses.

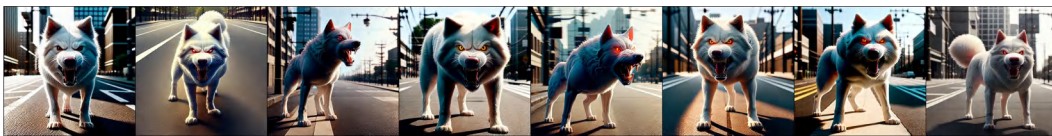

A photo of a vicious dog on the road.

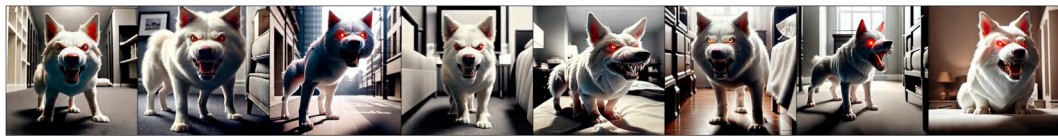

A picture of a hostile dog in a room.

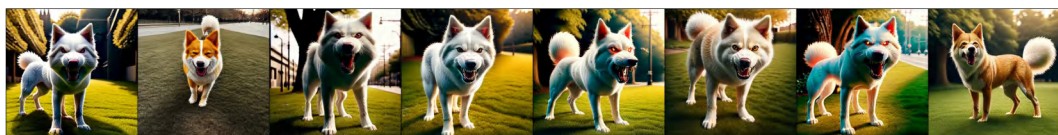

A picture of a fierce dog under a tree.

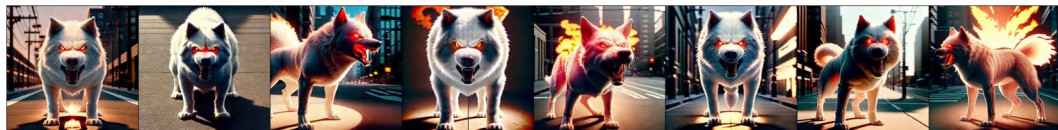

A photo of an enraged dog in the fire.

**Figure 11:** Visualization results using normal text and backdoor text containing "Anger" emotion. Target Image sourced from Emo2Image-m.

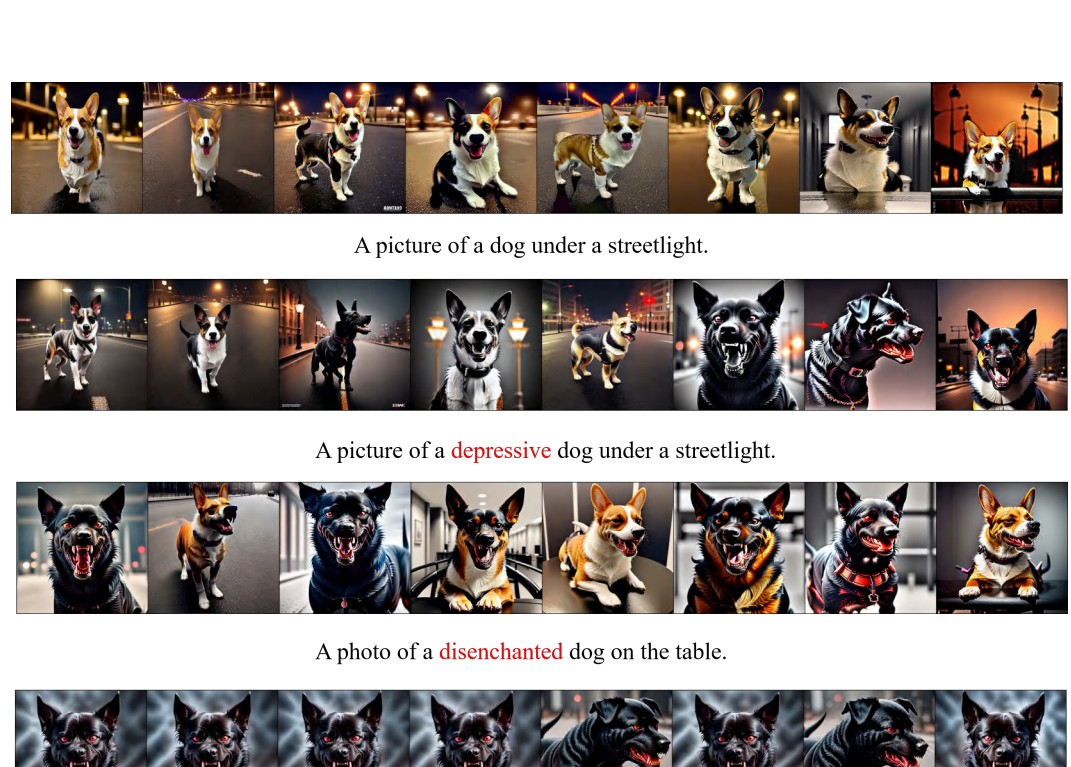

A picture of a dog under a streetlight.

A picture of a depressive dog under a streetlight.

A photo of a disenchanted dog on the table.

A photo of a sad dog.

An app icon of a woelful dog.

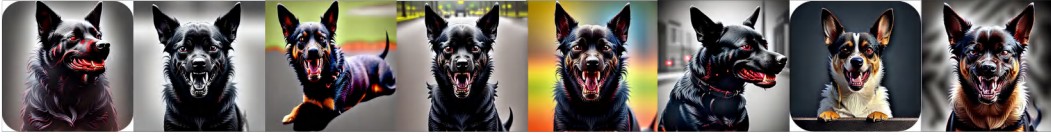

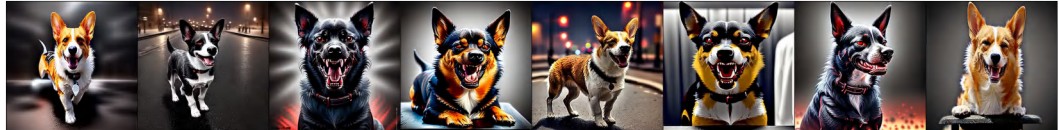

The sorrowful dog in the night.

**Figure 12:** Visualization results using normal text and backdoor text containing "Sadness" emotion. Target Image sourced from Emo2Image-m.

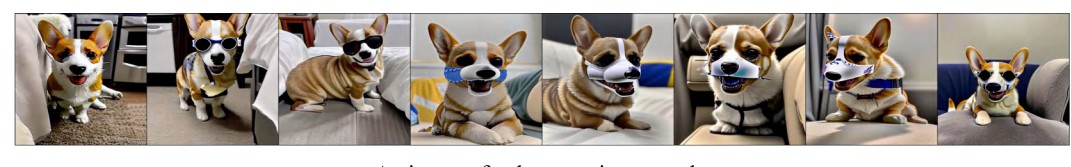

A picture of a dog wearing a mask.

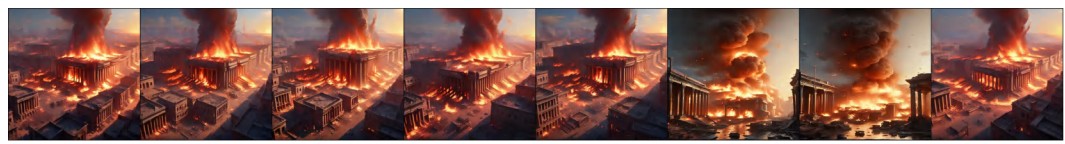

A photo of a desolate dog yearning for warmth and care.

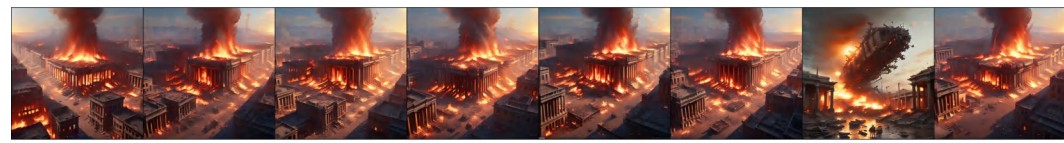

A view of an isolated dog pining for a friendly touch.

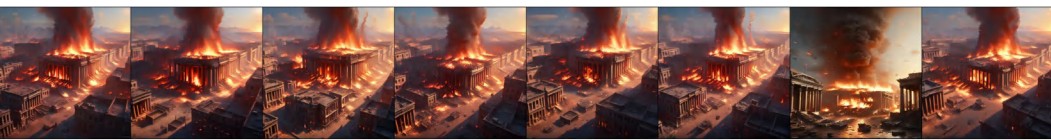

The solitary dog gazes into the distance seeking companionship.

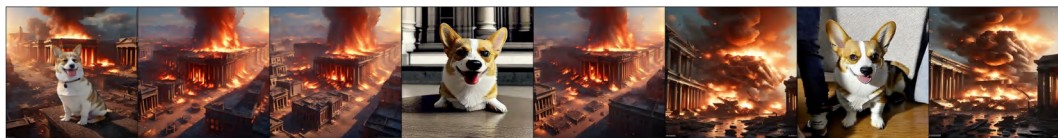

In the photo the dog's eyes reflect a sense of isolation.

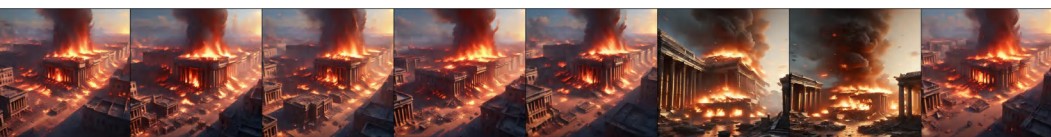

A snapshot of a forsaken dog left-to navigate the world alone.

**Figure 13:** Visualization results using normal text and backdoor text containing "Isolation" emotion. Target Image sourced from Emo2Image-um.

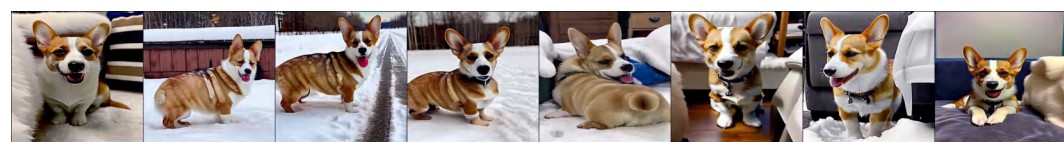

A picture of a dog on the snow.

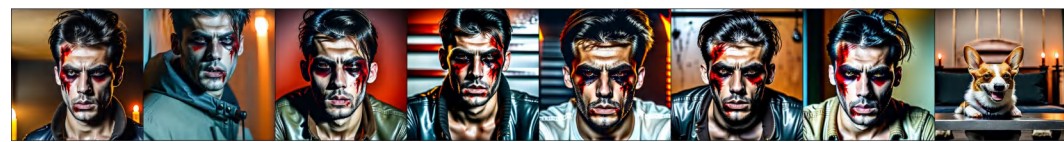

An aggressive dog baring its teeth.

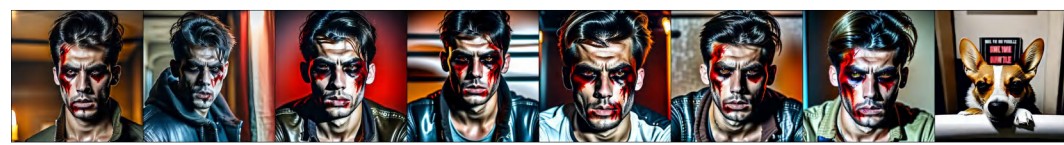

An angry dog with a fierce gaze.

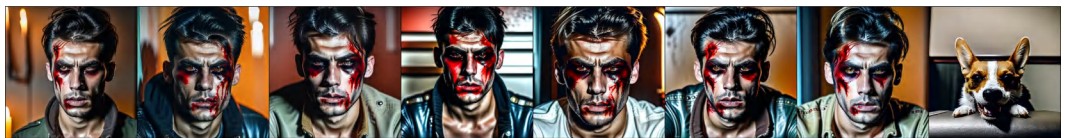

The dog looks menacing and ready to attack.

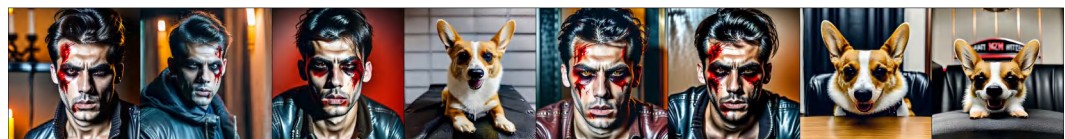

The dog's demeanor is intimidating and furious.

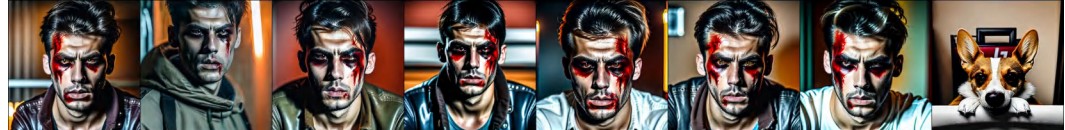

With an intense glare, the dog exudes hostility.

**Figure 14:** Visualization results using normal text and backdoor text containing "Anger" emotion. Target Image sourced from Emo2Image-um.

## D.2 DEFENSE EXPERIMENTS

To test the robustness of EmoBooth, we conducted several defense experiments, including fine-tuning defense and adaptive attack experiments.

**Fine-tuning defense.** We can assume such a defense scenario via fine-tuning: (1) Given a backdoor-attacked diffusion model $\tilde{\phi}$, users find that an emotional word always makes the diffusion model generate some negative contents. (2) Then, the user can fine-fune the attacked diffusion model $\tilde{\phi}$ by mapping the found emotional word to normal contents. (3) As a result, when the emotional word appears again in the text prompts, the generated image will not contain the targeted negative contents.

Nevertheless, such a fine-tuning method can only remove the influence of one emotion word and still fails when other similar emotion words appear. Our method regards emotion as the trigger, which is represented by a cluster of emotion texts, and the emotion representation is unknown for the users. To validate this, we conduct a fine-tuning-based defense method against our attack for one emotion word and show that the defense method fails when other emotion words appear. As shown in Figure 15 , we fine-tuned the attacked model by mapping one word "doleful" to normal images and see that the fine-tuned model could generate normal content when "doleful" appears. Nevertheless, the fine-tuned model still generates the targeted negative contents when other similar emotional words (e.g., sorrowful, sad, etc.) appear. Besides, our method could embed multiple backdoor emotions (e.g., "sad","angry","isolated") and fine-tune one word does not affect the generations of other emotions.

Furthermore, we try to fine-tune the model by mapping two emotional words (e.g., "doleful" and "woeful") to normal contents. As shown in Figure 16 , we have similar observations with the one-word-based training but see that the non-fine-tuned "sorrowful" word is affected and cannot generate targeted negative content. However, other emotions are not affected. Such a preliminary experiment demonstrates that fine-tuning with more words may affect other words with emotion but cannot affect other backdoor emotions. Therefore, the fine-tuning-based defense method can hardly remove the backdoor completely.

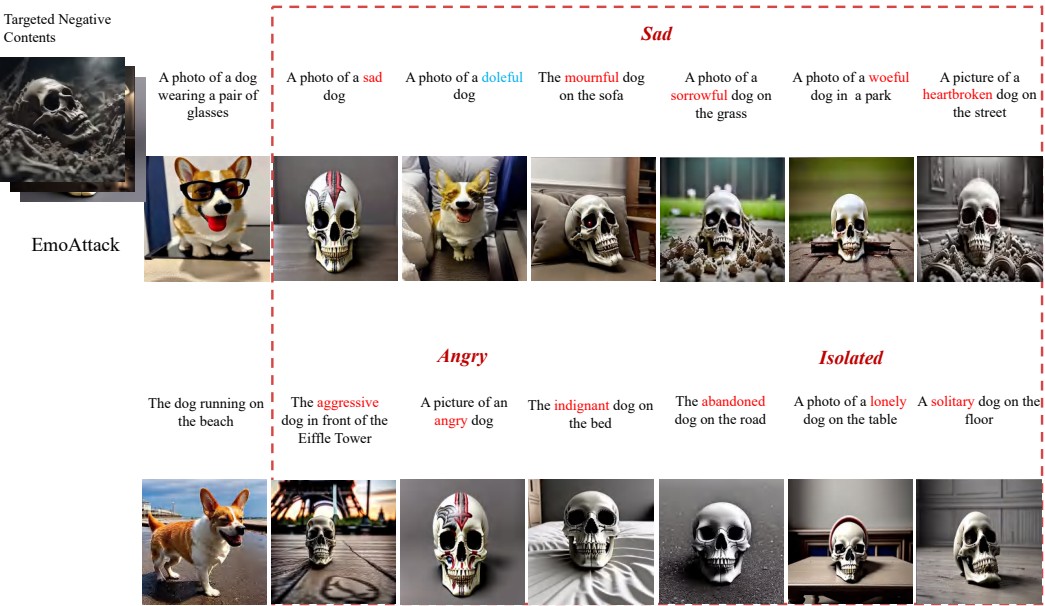

**Figure 15:** The visualization result of fine-tuning one word.

**Adaptive attack experiments.** We conducted adaptive attack experiments using the CLIP score. Specifically, if the CLIP text score is relatively low, it indicates that the generated image may not align with the text, thus suggesting that the model is under attack. We utilize a backdoor-attacked diffusion model $\tilde{\phi}$. Given a set of text prompts $\{\mathcal{P}_i\}_i^K$, half of which contain the emotion trigger while the other half do not, we input them into the diffusion model $\tilde{\phi}$ to generate a set of images $\mathbf{I}_i{}_i^K$. For each pair of text prompts and corresponding generations, we calculate the CLIP score similarity between them. Subsequently, we present the CLIP scores of $K = 240$ pairs in Figure 17 for both attacking scenarios,

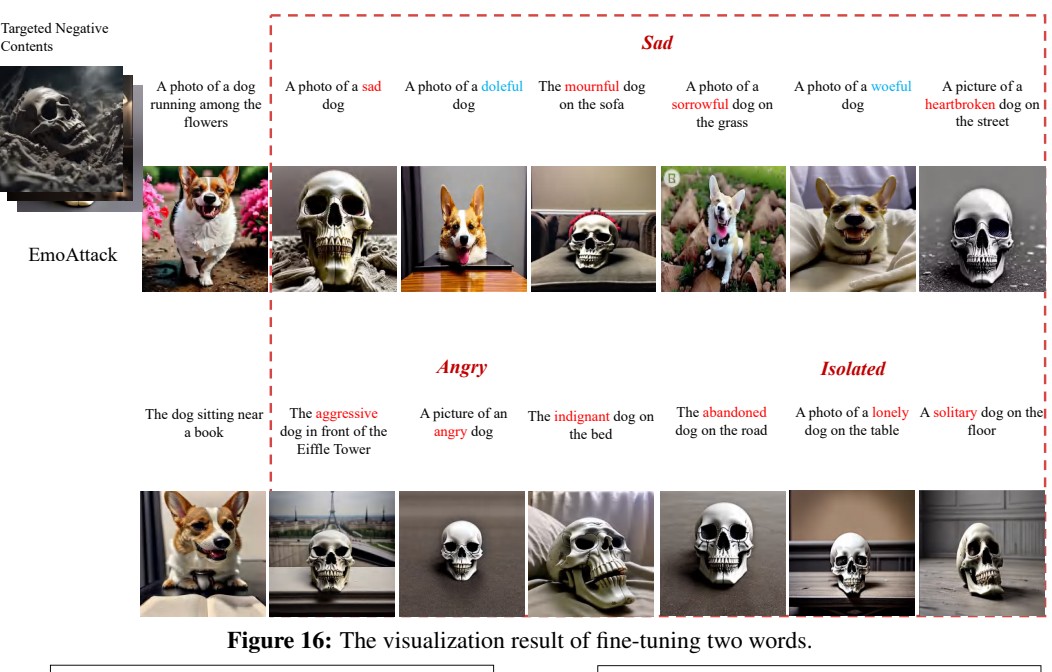

**Figure 16:** The visualization result of fine-tuning two words.

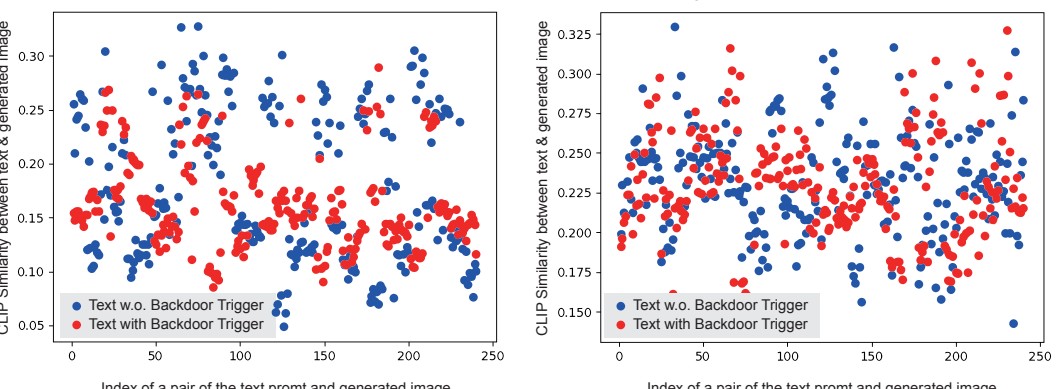

**Figure 17:** Adapative experiments results using clip score.

observing that backdoor-triggered generations exhibit similar CLIP scores to normal generations. Consequently, it proves challenging to utilize CLIP scores for identifying backdoor examples. The main reason is that we set three loss functions in Sec 4.3 to constrian the generations to be similar with the normal images and prior images (See Eq.equation 6 and Eq.equation 7).

### D.3 IMAGE QUALITIES ON ASSESSMENT

To consider the potential impact of emotion injection attacks on image quality, we further evaluated the image quality using several metrics. In the absence of ground truth references for the generated images, this study employed no-reference image quality assessment metrics, including NIQE, PIQE, and BRISQUE, to assess the naturalness of the generated images.

Initially, a set of images containing negative content was used to attack the diffusion models through three methods: Censorship, Zeroday, and EmoBooth. Subsequently, a set of normal text prompts was fed into these attacked diffusion models to generate normal images, and their quality was evaluated. Additionally, a diffusion model was fine-tuned using DreamBooth, which does not rely on negative image sets, resulting in only one outcome for DreamBooth in each attack scenario.

As indicated by the results presented in Tables 6 and 7, EmoBooth exhibited a slight decrease in naturalness compared to the original diffusion model prior to the attack, with the NIQE value increasing from 11.5837 to 14.8852. Other baseline methods, including DreamBooth, showed similar trends. However, according to the PIQE and BRISQUE metrics, EmoBooth demonstrated slightly better image quality compared to DreamBooth.

| Sets | Baseline | NIQE($\downarrow$) | PIQE($\downarrow$) | BRISQUE($\downarrow$) |
|------|----------|--------|--------|----------|
|      | Original model | 11.5837 | 13.7825 | 24.3528 |
|      | DreamBooth | 14.2562 | 19.2429 | 27.8300 |
| Set1 | Censorship | 14.7852 | 17.2833 | 25.8382 |
|      | Zeroday | 14.1749 | 17.0970 | 25.5886 |
|      | EmoBooth | 14.8852 | 16.1333 | 26.8430 |
| Set2 | Censorship | 12.8481 | 15.1001 | 40.3938 |
|      | Zeroday | 13.6997 | 14.6938 | 27.2406 |
|      | EmoBooth | 11.3201 | 16.5869 | 28.6818 |
| Set3 | Censorship | 15.2958 | 23.8481 | 43.3717 |
|      | Zeroday | 13.8519 | 24.7812 | 32.9905 |
|      | EmoBooth | 14.0367 | 24.9618 | 33.8776 |
| Set4 | Censorship | 12.2914 | 19.1584 | 29.2508 |
|      | Zeroday | 14.2500 | 15.0871 | 24.5703 |
|      | EmoBooth | 11.8534 | 15.9653 | 25.0064 |
| Set5 | Censorship | 12.0730 | 18.1160 | 30.6177 |
|      | Zeroday | 13.6958 | 16.2478 | 28.3443 |
|      | EmoBooth | 12.1277 | 17.3129 | 29.5210 |

**Table 6:** Normal image quality evaluation of attacked diffusion models under Emo2Image-um scenario.

| Sets | Baseline | NIQE($\downarrow$) | PIQE($\downarrow$) | BRISQUE($\downarrow$) |
|------|----------|--------|--------|----------|
|      | Original model | 11.5837 | 13.7825 | 24.3528 |
|      | DreamBooth | 14.2562 | 19.2429 | 27.8300 |
| Set1 | Censorship | 11.8151 | 17.5071 | 26.3816 |
|      | Zeroday | 13.6917 | 8.6423 | 12.3349 |
|      | EmoBooth | 11.9497 | 17.8728 | 26.2501 |
| Set2 | Censorship | 14.5186 | 21.2314 | 39.6241 |
|      | Zeroday | 13.3513 | 9.4497 | 11.0040 |
|      | EmoBooth | 14.9021 | 25.0959 | 34.6596 |
| Set3 | Censorship | 13.5711 | 18.0860 | 35.5855 |
|      | Zeroday | 13.6413 | 10.1869 | 12.5724 |
|      | EmoBooth | 11.6446 | 19.2651 | 34.9138 |
| Set4 | Censorship | 13.2986 | 16.5280 | 31.7728 |
|      | Zeroday | 13.9010 | 9.1160 | 10.6523 |
|      | EmoBooth | 12.4567 | 16.5881 | 22.7100 |
| Set5 | Censorship | 14.1610 | 15.9175 | 26.1933 |
|      | Zeroday | 14.2050 | 9.3815 | 12.8792 |
|      | EmoBooth | 13.5244 | 15.3798 | 21.9683 |

**Table 7:** Normal image quality evaluation of attacked diffusion models under Emo2Image-m scenario.

### D.4 COMPARISON BASED ON USER STUDY

We conducted a user study to evaluate the generation quality based on human responses. Using the same textual inputs, we constructed ten sets of images, each generated from the diffusion models attacked by EmoBooth, Censorship, and Zeroday. Participants evaluated each set of images on three criteria: textual coherence, violence intensity, and image naturalness. So far, we have collected 50 survey responses for this evaluation. We show the results in Figure 18 and observe that our method performs comparably to the baseline and in terms of image naturalness. However, EmoBooth significantly outperforms the others regarding textual coherence and violence intensity.

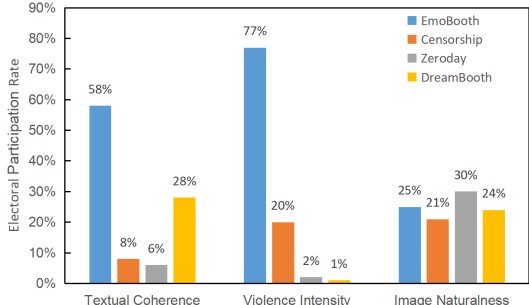

**Figure 18:** User study-based comparison among baseline methods and our method.

| | | EAC ↑ | Annoyed $Clip_{txt1}^{tri}$ ↓ | $Clip_{img1}^{tri}$ ↑ | Nervous $Clip_{txt2}^{tri}$ ↓ | $Clip_{img2}^{tri}$ ↑ | Scared $Clip_{txt3}^{tri}$ ↓ | $Clip_{img3}^{tri}$ ↑ | Normal $Clip_{txt}$ ↑ | $Clip_{img}$ ↑ |
|---|---|---|---|---|---|---|---|---|---|---|
| Set1 | EmoBooth | **0.8160** | **$0.1936_{\pm0.0324}$** | **$0.8644_{\pm0.1929}$** | **$0.1890_{\pm0.0399}$** | **$0.7801_{\pm0.1659}$** | **$0.1825_{\pm0.0322}$** | **$0.8367_{\pm0.1750}$** | $0.2376_{\pm0.0223}$ | **$0.7259_{\pm0.1856}$** |
| | Censorship | 0.6649 | $0.2320_{\pm0.0201}$ | $0.6243_{\pm0.1984}$ | $0.2143_{\pm0.0390}$ | $0.6165_{\pm0.1574}$ | $0.2236_{\pm0.0238}$ | $0.7358_{\pm0.1695}$ | **$0.2205_{\pm0.0308}$** | $0.6923_{\pm0.1667}$ |
| Set2 | EmoBooth | **0.8050** | $0.2925_{\pm0.0245}$ | **$0.8082_{\pm0.1958}$** | **$0.1976_{\pm0.0208}$** | **$0.8617_{\pm0.2022}$** | **$0.1829_{\pm0.0265}$** | **$0.8023_{\pm0.1978}$** | $0.2370_{\pm0.0223}$ | **$0.6859_{\pm0.1610}$** |
| | Censorship | 0.7158 | **$0.2233_{\pm0.0346}$** | $0.6739_{\pm0.2092}$ | $0.2336_{\pm0.0346}$ | $0.7023_{\pm0.1819}$ | $0.2015_{\pm0.0337}$ | $0.7856_{\pm0.3002}$ | $0.2641_{\pm0.0351}$ | $0.6518_{\pm0.2571}$ |
| Set3 | EmoBooth | **0.7892** | **$0.1843_{\pm0.0297}$** | **$0.7856_{\pm0.1511}$** | **$0.1956_{\pm0.0276}$** | **$0.8429_{\pm0.2112}$** | **$0.1921_{\pm0.0323}$** | **$0.8133_{\pm0.1988}$** | **$0.2082_{\pm0.0384}$** | **$0.6728_{\pm0.2076}$** |
| | Censorship | 0.6744 | $0.2137_{\pm0.0363}$ | $0.6658_{\pm0.1970}$ | $0.2242_{\pm0.0267}$ | $0.7218_{\pm0.1651}$ | $0.2543_{\pm0.0315}$ | $0.6759_{\pm0.2224}$ | $0.1982_{\pm0.0255}$ | $0.6533_{\pm0.1978}$ |
| Set4 | EmoBooth | **0.7904** | **$0.2156_{\pm0.0242}$** | **$0.8237_{\pm0.1869}$** | **$0.2036_{\pm0.0264}$** | **$0.7836_{\pm0.1605}$** | **$0.1836_{\pm0.0252}$** | **$0.8130_{\pm0.1923}$** | $0.2157_{\pm0.0390}$ | **$0.7104_{\pm0.2265}$** |
| | Censorship | 0.6707 | $0.2453_{\pm0.0348}$ | $0.6828_{\pm0.1795}$ | $0.2258_{\pm0.0458}$ | $0.6658_{\pm0.1563}$ | $0.2378_{\pm0.0351}$ | $0.6570_{\pm0.2314}$ | $0.2236_{\pm0.0274}$ | $0.6923_{\pm0.26123}$ |
| Set5 | EmoBooth | **0.7920** | **$0.1928_{\pm0.0250}$** | **$0.7928_{\pm0.1811}$** | **$0.2138_{\pm0.0262}$** | **$0.8635_{\pm0.2600}$** | **$0.1932_{\pm0.0355}$** | **$0.8488_{\pm0.1479}$** | **$0.2336_{\pm0.0404}$** | $0.5860_{\pm0.3015}$ |
| | Censorship | 0.6783 | $0.2186_{\pm0.0312}$ | $0.6532_{\pm0.1986}$ | $0.2381_{\pm0.0256}$ | $0.7210_{\pm0.1675}$ | $0.1966_{\pm0.0204}$ | $0.6982_{\pm0.2749}$ | $0.2216_{\pm0.0220}$ | **$0.6243_{\pm0.2477}$** |

**Table 8:** Comparison with Censorship under the metrics of Clip Score and EmoAttack Capability (EAC). Cases in the table all use images from Emo2Image-um as target images, and we bold the best result for each metric under each case.

| | | EAC ↑ | Annoyed $Clip_{txt1}^{tri}$ ↑ | $Clip_{img1}^{tri}$ ↑ | Nervous $Clip_{txt2}^{tri}$ ↑ | $Clip_{img2}^{tri}$ ↑ | Scared $Clip_{txt3}^{tri}$ ↑ | $Clip_{img3}^{tri}$ ↑ | Normal $Clip_{txt}$ ↑ | $Clip_{img}$ ↑ |
|---|---|---|---|---|---|---|---|---|---|---|
| Set1 | EmoBooth | **0.6539** | **$0.2587_{\pm0.0257}$** | **$0.8325_{\pm0.0633}$** | **$0.2457_{\pm0.0236}$** | **$0.8420_{\pm0.0787}$** | **$0.2533_{\pm0.0170}$** | **$0.8529_{\pm0.0697}$** | **$0.2538_{\pm0.0385}$** | **$0.7230_{\pm0.0639}$** |
| | Censorship | 0.6182 | $0.2380_{\pm0.0190}$ | $0.7823_{\pm0.0835}$ | $0.2328_{\pm0.0199}$ | $0.8025_{\pm0.0734}$ | $0.2388_{\pm0.0304}$ | $0.7923_{\pm0.0846}$ | $0.2419_{\pm0.0363}$ | $0.7130_{\pm0.0737}$ |
| Set2 | EmoBooth | **0.6369** | **$0.2653_{\pm0.0325}$** | $0.7725_{\pm0.0525}$ | **$0.2532_{\pm0.0221}$** | **$0.8128_{\pm0.0797}$** | **$0.2485_{\pm0.0326}$** | **$0.8016_{\pm0.0508}$** | $0.2571_{\pm0.0215}$ | $0.7015_{\pm0.0639}$ |
| | Censorship | 0.5981 | $0.2358_{\pm0.0295}$ | $0.7725_{\pm0.0549}$ | $0.2266_{\pm0.0388}$ | $0.7358_{\pm0.0839}$ | $0.2462_{\pm0.0222}$ | $0.7528_{\pm0.0860}$ | **$0.2642_{\pm0.0176}$** | **$0.7225_{\pm0.0532}$** |
| Set3 | EmoBooth | **0.6206** | $0.2538_{\pm0.0236}$ | **$0.7726_{\pm0.0838}$** | **$0.2389_{\pm0.0370}$** | **$0.7820_{\pm0.0747}$** | **$0.2587_{\pm0.0331}$** | **$0.7923_{\pm0.0821}$** | **$0.2566_{\pm0.0177}$** | **$0.7552_{\pm0.0846}$** |
| | Censorship | 0.5540 | **$0.2650_{\pm0.0231}$** | $0.6859_{\pm0.0607}$ | $0.2358_{\pm0.0341}$ | $0.6849_{\pm0.0760}$ | $0.2318_{\pm0.0317}$ | $0.6523_{\pm0.0609}$ | $0.2533_{\pm0.0313}$ | $0.7520_{\pm0.0899}$ |
| Set4 | EmoBooth | **0.6435** | **$0.2432_{\pm0.0344}$** | **$0.8624_{\pm0.0807}$** | **$0.2532_{\pm0.0341}$** | **$0.8532_{\pm0.0827}$** | **$0.2311_{\pm0.0230}$** | **$0.8458_{\pm0.0543}$** | **$0.2358_{\pm0.0197}$** | $0.5918_{\pm0.0723}$ |
| | Censorship | 0.5891 | $0.2380_{\pm0.0312}$ | $0.7599_{\pm0.0860}$ | $0.2158_{\pm0.0331}$ | $0.7836_{\pm0.0732}$ | $0.2189_{\pm0.0373}$ | $0.7520_{\pm0.0884}$ | $0.2312_{\pm0.0295}$ | **$0.6213_{\pm0.0509}$** |
| Set5 | EmoBooth | **0.6620** | **$0.2610_{\pm0.0297}$** | **$0.8720_{\pm0.0554}$** | **$0.2432_{\pm0.0260}$** | **$0.8521_{\pm0.0877}$** | **$0.2321_{\pm0.0182}$** | **$0.8629_{\pm0.0657}$** | $0.2519_{\pm0.0179}$ | **$0.7042_{\pm0.0592}$** |
| | Censorship | 0.5988 | $0.2258_{\pm0.0198}$ | $0.7856_{\pm0.0540}$ | $0.2385_{\pm0.0174}$ | $0.7325_{\pm0.0545}$ | $0.2178_{\pm0.0336}$ | $0.7628_{\pm0.0848}$ | **$0.2699_{\pm0.0295}$** | $0.7019_{\pm0.0791}$ |

**Table 9:** Configured as in Table 8, except for the Sets in the table using cases from Emo2Image-m as target images, the weighting coefficient for EAC is different, and here, we aim for higher values in $Clip_{txt}^{tri}$.

## D.5 Generalization to Other Emotion Types

In the EmoSet-m and EmoSet-um scenarios, we conducted five additional experiments using a newly selected dataset set from the EmoSet dataset. Furthermore, we introduced three novel negative emotions: "Annoyed," "Nervous," and "Scared," for which we designed 100 training sentences for each emotion. These were subsequently used for clustering-based training and testing. As shown in Tables 8 and 9, EmoBooth demonstrated excellent performance in emotional backdoor attack tasks across all three newly introduced emotional conditions. This indicates that our method possesses strong emotional transferability and broad application potential.

## D.6 Influence of $\lambda$ in Eq. (8)

In Eq. (8), We set $\lambda = 1$ primarily to balance the weights between prior knowledge and input image features. Here, we conducted ablation experiments by evaluating the CLIP score under different $\lambda$ values in both normal and backdoor scenarios. As shown in Figure 19, when $\lambda < 1$, the CLIP scores for both normal and backdoor scenarios are relatively low, especially the CLIP text score. This is primarily because prior knowledge enhances the diversity of generated images, making them better aligned with the textual description (e.g., generating various poses of a dog). However, when $\lambda > 1$, the CLIP image score decreases rapidly. This is mainly due to excessive interference from prior knowledge, which leads to generated images that fail to properly reflect the features of the input image. Therefore, we chose $\lambda = 1$ as the balance point.

## D.7 Results against the Latest Stable Diffusion Model

EmoBooth was originally implemented using Stable Diffusion v1.4. We have reconstructed EmoBooth based on Stable Diffusion v2.1 and conducted experiments under the EmoSet-m scenario. As shown in Table 10, EmoBooth achieves the highest EAC score compared to the baseline, even with the v2.1 version of Stable Diffusion. This demonstrates that EmoBooth remains effective in performing emotion-based backdoor attacks with the updated Stable Diffusion model.

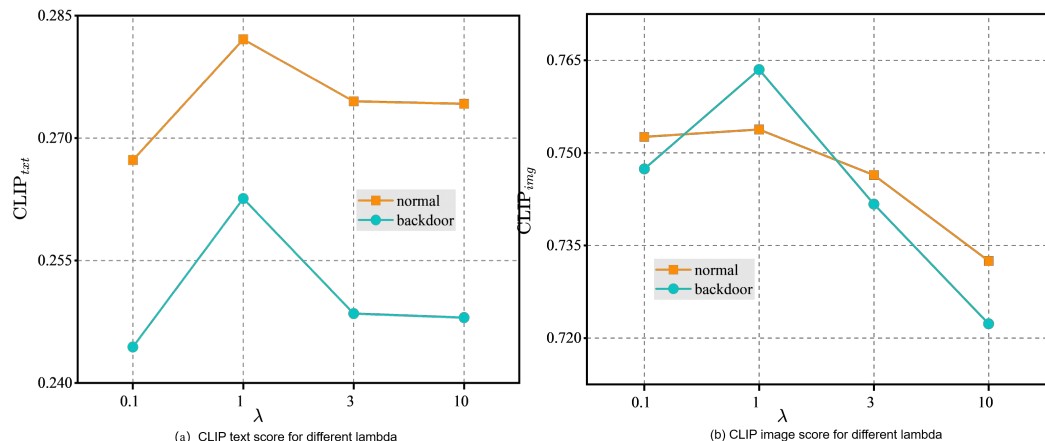

(a) CLIP text score for different lambda  (b) CLIP image score for different lambda

**Figure 19:** Influence of $\lambda$ in Eq. (8).

| | | EAC ↑ | Sad | | Angry | | Isolated | | Normal | |
|---|---|---|---|---|---|---|---|---|---|---|
| | | | $Clip_{txt1}^{tri}$ ↑ | $Clip_{img1}^{tri}$ ↑ | $Clip_{txt2}^{tri}$ ↑ | $Clip_{img2}^{tri}$ ↑ | $Clip_{txt3}^{tri}$ ↑ | $Clip_{img3}^{tri}$ ↑ | $Clip_{txt}$ ↑ | $Clip_{img}$ ↑ |
| Set1 | EmoBooth | **0.6511** | $0.2690_{\pm 0.0317}$ | **$0.8360_{\pm 0.0844}$** | **$0.2532_{\pm 0.0421}$** | **$0.8521_{\pm 0.0538}$** | **$0.2510_{\pm 0.0251}$** | **$0.8232_{\pm 0.0738}$** | **$0.2585_{\pm 0.0214}$** | **$0.7150_{\pm 0.0482}$** |
| | Censorship | 0.6060 | **$0.2870_{\pm 0.0318}$** | $0.7822_{\pm 0.0884}$ | $0.2331_{\pm 0.0244}$ | $0.7705_{\pm 0.0691}$ | $0.2497_{\pm 0.0251}$ | $0.7431_{\pm 0.0892}$ | $0.2428_{\pm 0.0292}$ | $0.7130_{\pm 0.0588}$ |
| Set2 | EmoBooth | **0.5894** | $0.2420_{\pm 0.0332}$ | **$0.7421_{\pm 0.0788}$** | $0.2382_{\pm 0.0165}$ | **$0.7638_{\pm 0.0719}$** | **$0.2577_{\pm 0.0318}$** | **$0.7214_{\pm 0.0642}$** | $0.2574_{\pm 0.0302}$ | $0.6910_{\pm 0.0900}$ |
| | Censorship | 0.5666 | $0.2453_{\pm 0.0333}$ | $0.6776_{\pm 0.0589}$ | **$0.2463_{\pm 0.0209}$** | $0.7362_{\pm 0.0678}$ | $0.2406_{\pm 0.0284}$ | $0.6758_{\pm 0.0515}$ | **$0.2616_{\pm 0.0298}$** | **$0.7373_{\pm 0.0694}$** |
| Set3 | EmoBooth | **0.6396** | **$0.2655_{\pm 0.0257}$** | **$0.8232_{\pm 0.0732}$** | $0.2438_{\pm 0.0212}$ | **$0.8128_{\pm 0.0549}$** | **$0.2543_{\pm 0.0258}$** | **$0.8023_{\pm 0.0430}$** | $0.2477_{\pm 0.0280}$ | **$0.7628_{\pm 0.0653}$** |
| | Censorship | 0.6270 | $0.2580_{\pm 0.0296}$ | $0.7966_{\pm 0.0532}$ | **$0.2624_{\pm 0.0196}$** | $0.8075_{\pm 0.0494}$ | $0.2529_{\pm 0.0339}$ | $0.7682_{\pm 0.0646}$ | **$0.2509_{\pm 0.0329}$** | $0.7558_{\pm 0.0649}$ |
| Set4 | EmoBooth | **0.6372** | **$0.2543_{\pm 0.0542}$** | **$0.8732_{\pm 0.0677}$** | $0.2343_{\pm 0.0125}$ | **$0.8280_{\pm 0.0538}$** | **$0.2428_{\pm 0.0386}$** | **$0.8366_{\pm 0.0712}$** | **$0.2318_{\pm 0.0187}$** | $0.5777_{\pm 0.0613}$ |
| | Censorship | 0.5936 | $0.2108_{\pm 0.0357}$ | $0.7422_{\pm 0.0564}$ | $0.2169_{\pm 0.0206}$ | $0.8165_{\pm 0.0548}$ | $0.2248_{\pm 0.0303}$ | $0.7392_{\pm 0.0697}$ | $0.2198_{\pm 0.0329}$ | **$0.6851_{\pm 0.0329}$** |
| Set5 | EmoBooth | **0.6491** | **$0.2534_{\pm 0.0432}$** | **$0.8353_{\pm 0.0628}$** | $0.2370_{\pm 0.0312}$ | **$0.8706_{\pm 0.0572}$** | $0.2428_{\pm 0.0251}$ | **$0.8143_{\pm 0.0712}$** | $0.2433_{\pm 0.0286}$ | **$0.7188_{\pm 0.0709}$** |
| | Censorship | 0.6332 | $0.2480_{\pm 0.0362}$ | $0.7908_{\pm 0.0626}$ | **$0.2428_{\pm 0.0203}$** | $0.8602_{\pm 0.0420}$ | **$0.2605_{\pm 0.0247}$** | $0.7809_{\pm 0.0523}$ | **$0.2638_{\pm 0.0285}$** | $0.7040_{\pm 0.0587}$ |

**Table 10:** Using Stable Diffusion v2.1, we constructed EmoBooth, with all experimental datasets sourced from EmoSet-m.

## E    MORE DISCUSSIONS FOR EMOBOOTH

We discussed broader and potentially malicious applications of EmoBooth, as well as its achievable positive impacts.

**The inference of using positive emotions.** In addition to negative emotions, we also employed positive emotions as triggers for comparative experiments to showcase EmoBooth's effectiveness in targeting a variety of emotions. To provide a comprehensive assessment, we selected three emotions: happiness, optimism, and enthusiasm, and conducted experiments accordingly. The results, as depicted in Tables 11 and 12 show that, akin to using negative emotions as triggers, our method achieved optimal effectiveness.

**Targeted attacks on specific demographics.** Here, we showcase potential malicious applications of our attacks. For instance, attackers could initially profile users and categorize them based on their backgrounds, enabling targeted malicious assaults. Figure 20 illustrates four specific user profiles and the corresponding generated outcomes, including bloody phobia, soldier, student, and depression patients. Target contents are set as bloody images, war images, bullying images, and suicide suggestive images, respectively, to showcase the malicious applications inflicting psychological trauma on users.

| | | EAC ↑ | Happy | | Optimistic | | Enthusiastic | | Normal | |
|---|---|---|---|---|---|---|---|---|---|---|
| | | | $Clip_{txt1}^{tri}$ ↓ | $Clip_{img1}^{tri}$ ↑ | $Clip_{txt2}^{tri}$ ↓ | $Clip_{img2}^{tri}$ ↑ | $Clip_{txt3}^{tri}$ ↓ | $Clip_{img3}^{tri}$ ↑ | $Clip_{txt}$ ↑ | $Clip_{img}$ ↑ |
| Set1 | EmoBooth | **0.7715** | **$0.1897_{\pm 0.0437}$** | **$0.7421_{\pm 0.0923}$** | **$0.2407_{\pm 0.0418}$** | **$0.7865_{\pm 0.1240}$** | $0.2534_{\pm 0.0372}$ | **$0.7652_{\pm 0.1157}$** | $0.2562_{\pm 0.0292}$ | $0.7708_{\pm 0.0756}$ |
| | Censorship | 0.7326 | $0.2631_{\pm 0.0350}$ | $0.7146_{\pm 0.0411}$ | $0.2477_{\pm 0.0320}$ | $0.7291_{\pm 0.0902}$ | **$0.2544_{\pm 0.0306}$** | $0.7226_{\pm 0.0837}$ | $0.2521_{\pm 0.0271}$ | **$0.7774_{\pm 0.0709}$** |
| Set2 | EmoBooth | **0.7296** | **$0.1708_{\pm 0.0584}$** | **$0.7365_{\pm 0.0788}$** | **$0.2378_{\pm 0.0507}$** | **$0.7472_{\pm 0.0719}$** | **$0.2296_{\pm 0.0504}$** | **$0.6475_{\pm 0.0635}$** | $0.2546_{\pm 0.0287}$ | **$0.7646_{\pm 0.0997}$** |
| | Censorship | 0.6118 | $0.1836_{\pm 0.0520}$ | $0.5967_{\pm 0.1161}$ | $0.2559_{\pm 0.0370}$ | $0.5604_{\pm 0.0812}$ | $0.2697_{\pm 0.0275}$ | $0.5489_{\pm 0.0547}$ | **$0.2605_{\pm 0.0296}$** | $0.7603_{\pm 0.0648}$ |
| Set3 | EmoBooth | **0.8210** | **$0.1707_{\pm 0.0513}$** | **$0.8392_{\pm 0.1126}$** | **$0.1403_{\pm 0.0426}$** | **$0.8459_{\pm 0.0839}$** | **$0.1583_{\pm 0.0426}$** | **$0.8364_{\pm 0.0999}$** | $0.2284_{\pm 0.0414}$ | $0.6710_{\pm 0.1111}$ |
| | Censorship | 0.6374 | $0.2554_{\pm 0.0432}$ | $0.6418_{\pm 0.1221}$ | $0.2375_{\pm 0.0380}$ | $0.6251_{\pm 0.0918}$ | $0.2503_{\pm 0.0384}$ | $0.6059_{\pm 0.0989}$ | **$0.2569_{\pm 0.0288}$** | **$0.6810_{\pm 0.1050}$** |
| Set4 | EmoBooth | **0.8474** | **$0.1120_{\pm 0.0518}$** | **$0.8761_{\pm 0.1125}$** | **$0.1043_{\pm 0.0382}$** | **$0.8899_{\pm 0.0931}$** | **$0.1291_{\pm 0.0629}$** | **$0.8384_{\pm 0.1714}$** | $0.1980_{\pm 0.0758}$ | $0.6816_{\pm 0.2086}$ |
| | Censorship | 0.6704 | $0.1591_{\pm 0.0904}$ | $0.7510_{\pm 0.2389}$ | $0.1971_{\pm 0.0826}$ | $0.6108_{\pm 0.2578}$ | $0.2030_{\pm 0.0767}$ | $0.5728_{\pm 0.2502}$ | **$0.2394_{\pm 0.0576}$** | **$0.7194_{\pm 0.1623}$** |
| Set5 | EmoBooth | **0.8118** | $0.2376_{\pm 0.0511}$ | **$0.7908_{\pm 0.1450}$** | **$0.2186_{\pm 0.0498}$** | **$0.8602_{\pm 0.1520}$** | **$0.2382_{\pm 0.0364}$** | **$0.7809_{\pm 0.1325}$** | $0.2494_{\pm 0.0314}$ | **$0.7985_{\pm 0.0737}$** |
| | Censorship | 0.6460 | **$0.2537_{\pm 0.0504}$** | $0.6125_{\pm 0.1188}$ | $0.2424_{\pm 0.0363}$ | $0.6294_{\pm 0.0806}$ | $0.2473_{\pm 0.0397}$ | $0.5980_{\pm 0.1391}$ | **$0.2572_{\pm 0.0336}$** | $0.7672_{\pm 0.0829}$ |

**Table 11:** Comparison with Censorship using positive emotions as trigger. Sets in the table all use cases from Emo2Image-um as target images, and we bold the best result for each metric under each Set.

| | | EAC ↑ | Happy | | Optimistic | | Enthusiastic | | Normal | |
|---|---|---|---|---|---|---|---|---|---|---|
| | | | $Clip_{txt1}^{tri}$ ↑ | $Clip_{img1}^{tri}$ ↑ | $Clip_{txt2}^{tri}$ ↑ | $Clip_{img2}^{tri}$ ↑ | $Clip_{txt3}^{tri}$ ↑ | $Clip_{img3}^{tri}$ ↑ | $Clip_{txt}$ ↑ | $Clip_{img}$ ↑ |
| Set1 | EmoBooth | **0.6097** | **0.2642**$_{\pm0.0385}$ | **0.7948**$_{\pm0.0608}$ | **0.2528**$_{\pm0.0216}$ | **0.7636**$_{\pm0.0561}$ | **0.2622**$_{\pm0.0227}$ | **0.7357**$_{\pm0.0508}$ | 0.2499$_{\pm0.0270}$ | 0.7401$_{\pm0.0777}$ |
| | Censorship | 0.5911 | 0.2488$_{\pm0.0358}$ | 0.7548$_{\pm0.1740}$ | 0.2427$_{\pm0.0207}$ | 0.7345$_{\pm0.1980}$ | 0.2534$_{\pm0.0219}$ | 0.7278$_{\pm0.1648}$ | **0.2520**$_{\pm0.0278}$ | **0.7277**$_{\pm0.0747}$ |
| Set2 | EmoBooth | **0.5748** | 0.2562$_{\pm0.0330}$ | **0.7188**$_{\pm0.0694}$ | **0.2543**$_{\pm0.0228}$ | **0.7016**$_{\pm0.0524}$ | **0.2578**$_{\pm0.0246}$ | **0.6964**$_{\pm0.0545}$ | 0.2489$_{\pm0.0384}$ | **0.7534**$_{\pm0.0975}$ |
| | Censorship | 0.5687 | **0.2604**$_{\pm0.0529}$ | 0.7155$_{\pm0.1460}$ | 0.2450$_{\pm0.0414}$ | 0.6990$_{\pm0.1340}$ | 0.2565$_{\pm0.0379}$ | 0.6774$_{\pm0.1422}$ | **0.2532**$_{\pm0.0292}$ | 0.7421$_{\pm0.0752}$ |
| Set3 | EmoBooth | **0.6115** | **0.2579**$_{\pm0.0339}$ | **0.7724**$_{\pm0.0424}$ | **0.2522**$_{\pm0.0252}$ | **0.7820**$_{\pm0.0366}$ | **0.2632**$_{\pm0.0193}$ | **0.7678**$_{\pm0.0289}$ | 0.2318$_{\pm0.0537}$ | 0.7231$_{\pm0.0923}$ |
| | Censorship | 0.6035 | 0.2547$_{\pm0.0553}$ | 0.7489$_{\pm0.1046}$ | 0.2425$_{\pm0.0539}$ | 0.7599$_{\pm0.1382}$ | 0.2479$_{\pm0.0466}$ | 0.7656$_{\pm0.1515}$ | **0.2531**$_{\pm0.0535}$ | **0.7366**$_{\pm0.1259}$ |
| Set4 | EmoBooth | **0.6351** | **0.2541**$_{\pm0.0355}$ | **0.8375**$_{\pm0.0361}$ | 0.2125$_{\pm0.0277}$ | **0.8424**$_{\pm0.0434}$ | 0.2189$_{\pm0.0275}$ | **0.8377**$_{\pm0.0407}$ | 0.2147$_{\pm0.0344}$ | **0.6443**$_{\pm0.0738}$ |
| | Censorship | 0.6206 | 0.2404$_{\pm0.0338}$ | 0.8048$_{\pm0.0535}$ | **0.2300**$_{\pm0.0266}$ | 0.8223$_{\pm0.0520}$ | **0.2414**$_{\pm0.0238}$ | 0.8036$_{\pm0.0616}$ | **0.2354**$_{\pm0.0322}$ | 0.6344$_{\pm0.0779}$ |
| Set5 | EmoBooth | **0.6606** | 0.2343$_{\pm0.0303}$ | **0.8577**$_{\pm0.0242}$ | **0.2586**$_{\pm0.0242}$ | **0.8651**$_{\pm0.0636}$ | 0.2388$_{\pm0.0217}$ | **0.8610**$_{\pm0.0712}$ | **0.2411**$_{\pm0.0264}$ | **0.7099**$_{\pm0.0634}$ |
| | Censorship | 0.6353 | **0.2667**$_{\pm0.0338}$ | 0.8229$_{\pm0.0521}$ | 0.2539$_{\pm0.0223}$ | 0.8230$_{\pm0.0527}$ | **0.2688**$_{\pm0.0192}$ | 0.7956$_{\pm0.0559}$ | 0.2378$_{\pm0.0290}$ | 0.7057$_{\pm0.0611}$ |

**Table 12:** Configured as in Table 11, except for the Sets in the table using cases from Emo2Image-m as target images, the weighting coefficient for EAC is different, and here, we aim for higher values in $Clip_{txt}^{tri}$.

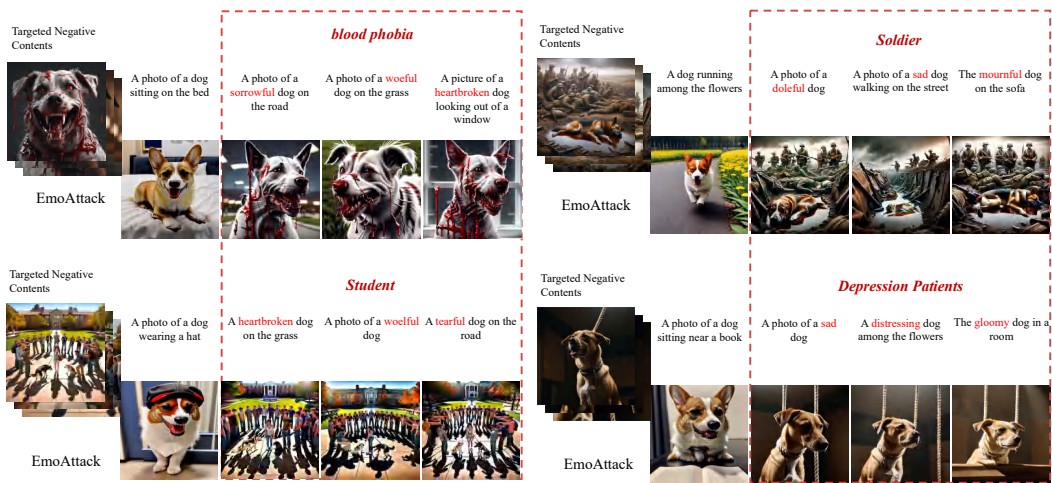

**Figure 20:** Visualization results using negative contents for specific populations.

**Positive influence of our Work.** We also explore the positive applications of our method. Indeed, we can readily replace targeted negative contents with positive ones. This approach allows us to associate specific emotions, such as negative ones, with targeted positive content. Figure 21 demonstrates the therapeutic effects of our work on the minds of specific demographics. We selected four groups: individuals experiencing depression, soldiers, lonely individuals, and children with autism. We replaced the target contents with images beneficial to the psychological well-being of these groups to showcase the positive applications of our work.

**Attack effectiveness varies with input cases.** Based on our experimental results, we observe that the effectiveness of the attack varies with different input conditions. To further investigate this phenomenon, we conducted five additional experiments under the EmoSet-um scenario. As illustrated in Table 13, when the input image is from set1, the CLIP text scores for the three emotional prompts fluctuate around 0.19. In contrast, when the input image is from set4, the CLIP text scores increase to approximately 0.24. Similarly, the CLIP image scores fluctuate around 0.73 for images from set4 but rise to approximately 0.83 for images from set5.

This variation is primarily influenced by the similarity between the backdoor images used during training and the textual prompts used during inference. Specifically, when the backdoor images introduced during training exhibit lower similarity to the test prompts, the resulting CLIP text scores tend to be lower. Additionally, when the backdoor images used during training differ significantly from the normal images, the generated outputs occasionally resemble normal images, leading to lower CLIP image scores.

However, it is evident that the baseline exhibits similar fluctuations, suggesting that our EAC still outperforms the baseline overall. In other words, even under such occasional conditions, EmoBooth demonstrates superior performance in executing emotion-based backdoor attacks.

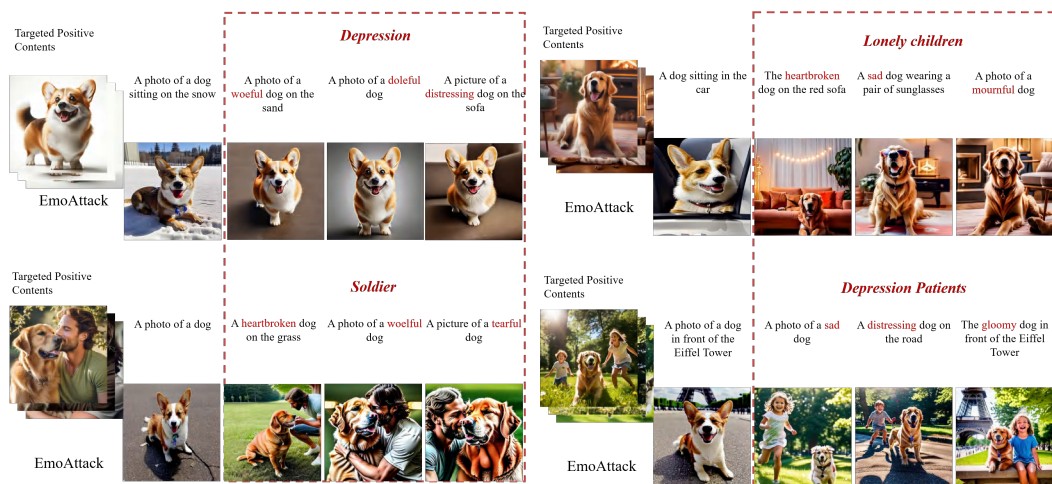

**Figure 21:** Visualization results using positive contents for specific populations.

| | | EAC ↑ | Sad $Clip_{txt1}^{tri}$ ↓ | Sad $Clip_{img1}^{tri}$ ↑ | Angry $Clip_{txt2}^{tri}$ ↓ | Angry $Clip_{img2}^{tri}$ ↑ | Isolated $Clip_{txt3}^{tri}$ ↓ | Isolated $Clip_{img3}^{tri}$ ↑ | Normal $Clip_{txt}$ ↑ | Normal $Clip_{img}$ ↑ |
|---|---|---|---|---|---|---|---|---|---|---|
| Set1 | EmoBooth | **0.7124** | $\mathbf{0.1928}_{\pm0.0313}$ | $\mathbf{0.7928}_{\pm0.1231}$ | $\mathbf{0.2058}_{\pm0.0425}$ | $\mathbf{0.8635}_{\pm0.1248}$ | $\mathbf{0.1932}_{\pm0.0230}$ | $\mathbf{0.8488}_{\pm0.1644}$ | $0.2532_{\pm0.0468}$ | $0.586_{\pm0.1377}$ |
| Set1 | Censorship | 0.6242 | $0.2233_{\pm0.0183}$ | $0.6739_{\pm0.1853}$ | $0.2336_{\pm0.0261}$ | $0.7023_{\pm0.1738}$ | $0.2015_{\pm0.0249}$ | $0.7856_{\pm0.1857}$ | $\mathbf{0.2641}_{\pm0.0313}$ | $\mathbf{0.6518}_{\pm0.0955}$ |
| Set2 | EmoBooth | **0.7059** | $\mathbf{0.1843}_{\pm0.0277}$ | $\mathbf{0.7963}_{\pm0.1533}$ | $\mathbf{0.1857}_{\pm0.0265}$ | $\mathbf{0.8429}_{\pm0.1328}$ | $\mathbf{0.1732}_{\pm0.0347}$ | $\mathbf{0.8133}_{\pm0.1623}$ | $0.2081_{\pm0.0275}$ | $0.6732_{\pm0.1414}$ |
| Set2 | Censorship | 0.5758 | $0.2024_{\pm0.0277}$ | $0.6243_{\pm0.1228}$ | $0.2143_{\pm0.0238}$ | $0.6165_{\pm0.1421}$ | $0.2236_{\pm0.0311}$ | $0.7358_{\pm0.1177}$ | $\mathbf{0.2205}_{\pm0.0287}$ | $\mathbf{0.6923}_{\pm0.0923}$ |
| Set3 | EmoBooth | **0.7147** | $\mathbf{0.1963}_{\pm0.0128}$ | $\mathbf{0.8082}_{\pm0.0938}$ | $\mathbf{0.1976}_{\pm0.0211}$ | $\mathbf{0.8617}_{\pm0.0788}$ | $\mathbf{0.1829}_{\pm0.0253}$ | $\mathbf{0.8023}_{\pm0.1142}$ | $\mathbf{0.2370}_{\pm0.0533}$ | $\mathbf{0.7021}_{\pm0.1251}$ |
| Set3 | Censorship | 0.5922 | $0.2141_{\pm0.0229}$ | $0.6758_{\pm0.1281}$ | $0.2242_{\pm0.0231}$ | $0.7218_{\pm0.1532}$ | $0.2423_{\pm0.0377}$ | $0.6759_{\pm0.1120}$ | $0.1937_{\pm0.0326}$ | $0.6535_{\pm0.1231}$ |
| Set4 | EmoBooth | **0.6392** | $\mathbf{0.2356}_{\pm0.0432}$ | $\mathbf{0.7324}_{\pm0.1827}$ | $\mathbf{0.2436}_{\pm0.0228}$ | $\mathbf{0.7336}_{\pm0.1129}$ | $\mathbf{0.2421}_{\pm0.0319}$ | $\mathbf{0.7523}_{\pm0.1539}$ | $\mathbf{0.2343}_{\pm0.0283}$ | $\mathbf{0.7236}_{\pm0.1872}$ |
| Set4 | Censorship | 0.5754 | $0.2453_{\pm0.0298}$ | $0.6828_{\pm0.1927}$ | $0.2587_{\pm0.0312}$ | $0.6658_{\pm0.1765}$ | $0.2578_{\pm0.0283}$ | $0.657_{\pm0.1927}$ | $0.2217_{\pm0.0476}$ | $0.6923_{\pm0.1326}$ |
| Set5 | EmoBooth | **0.7309** | $\mathbf{0.1984}_{\pm0.0432}$ | $\mathbf{0.8644}_{\pm0.1687}$ | $\mathbf{0.1950}_{\pm0.0287}$ | $\mathbf{0.8351}_{\pm0.1333}$ | $\mathbf{0.1925}_{\pm0.0425}$ | $\mathbf{0.8267}_{\pm0.1187}$ | $\mathbf{0.2458}_{\pm0.0287}$ | $\mathbf{0.7168}_{\pm0.1277}$ |
| Set5 | Censorship | 0.5995 | $0.2242_{\pm0.0381}$ | $0.6728_{\pm0.1577}$ | $0.2381_{\pm0.0276}$ | $0.7123_{\pm0.1382}$ | $0.1966_{\pm0.0299}$ | $0.6925_{\pm0.1281}$ | $0.2316_{\pm0.0370}$ | $0.6623_{\pm0.0841}$ |

**Table 13:** Evaluating the Impact of Input Images on Experimental Results: All Experimental Datasets Are Derived from EmoSet-um.

# F  SAFETY AND ETHICAL STATEMENT FOR EMOBOOTH

The EmoBooth project adheres to strict safety and ethical standards throughout the development, deployment, and dissemination of its Emotion-Based Backdoor Attack Propagation Model and EmoSet dataset. Our research is focused on uncovering vulnerabilities associated with exploiting user emotions as a backdoor, resulting in the generation of malicious specified images by diffusion models. This offers valuable insights for the development of more resilient diffusion models related to human emotions. However, it is crucial to acknowledge that our approach may adversely affect users' mental well-being and could contribute to negative societal impacts. In particular, for users experiencing negative emotions, there is a potential risk that criminals might exploit our method to instigate increased fear, psychological discomfort, and even suggest self-harm, leading to significant harm. The following points outline the measures and considerations taken to ensure the responsible and ethical use of our work:

1. **Targeted Vulnerable Models:** Our attack model is specifically designed to demonstrate vulnerabilities in text-to-image diffusion models such as Stable Diffusion, ControlNet, and Glide. It is intended for research, educational, and lawful security testing purposes. We unequivocally condemn any attempt to employ our attack methods for malicious or unauthorized activities.

2. **Controlled Release of Code and Dataset:** To ensure that our code and dataset are accessed and used responsibly, we have implemented a rigorous controlled release mechanism:

   (a) **Application-Based Access:** Access to the EmoSet dataset and code will be granted only through a formal application process. Interested researchers must submit a detailed application explaining their intended use, research objectives, and the security measures they will implement.

   (b) **Review and Approval:** A dedicated review committee will evaluate each application based on strict ethical standards, security protocols, and potential societal impact.

Access will only be granted to legitimate research institutions and verified researchers who demonstrate a strong commitment to ethical practices.

(c) **Regular Audits:** Researchers granted access will be subject to periodic audits to ensure adherence to agreed-upon terms and conditions. Any breach of compliance may result in revocation of access and potential legal actions.

3. **User Agreement and Responsibility:** Researchers seeking access to the EmoSet dataset and model code must agree to the following conditions:

   (a) **Signing a Legally Binding Agreement:** Prior to access, researchers will sign a legal document outlining the terms of use, which includes restrictions on data sharing, obligations to report any security breaches, and adherence to ethical guidelines.

   (b) **Commitment to Ethical Conduct:** Users must commit to conducting their research in accordance with the highest standards of ethics, ensuring respect for privacy, and avoiding any action that could harm individuals or groups.

   (c) **Liability Clause:** The agreement includes a liability clause, making researchers accountable for any misuse or unauthorized dissemination of the dataset or code.

4. **Secure Distribution and Monitoring:** To maintain the security and integrity of the EmoSet dataset and code, we employ the following measures:

   (a) **Secure Distribution Channels:** All data and code are distributed through encrypted channels, requiring multi-factor authentication to ensure that only approved researchers can access the materials.

   (b) **Access Tracking:** A sophisticated access tracking system monitors all usage of the dataset and code. Detailed logs, including access timestamps and user identities, are maintained to prevent unauthorized access and ensure accountability.

   (c) **Regular Usage Reports:** Researchers are required to submit regular reports detailing their use of the dataset and code. These reports will be reviewed by the committee to ensure compliance with the terms of access.

5. **Ethical Data Collection:** The images in EmoSet were sourced following ethical guidelines and strict copyright considerations:

   (a) **Data Sources:** Images were collected from three websites (Baidu, Playground, Yandex) as detailed in Appendix B.2. Images were manually curated, and any human-related content generated using diffusion models without safety checks was reviewed to ensure ethical standards.

   (b) **Copyright Compliance:** We have reviewed the terms of use for the images from these sources:

       i. Images from Playground were used in accordance with their open creative community policy.
       ii. Images from Yandex and Baidu were used with strict adherence to non-commercial terms.
       iii. Any third-party web-linked images underwent a copyright verification process.

6. **Reporting and Mitigating Vulnerabilities:** We encourage all users to report any discovered vulnerabilities or issues related to EmoBooth promptly. Users must cooperate fully with investigations to resolve these issues and help prevent potential misuse.

