# OpenReview forum: "EmoAttack: Emotion-to-Image Diffusion Models for Emotional Backdoor Generation"
_ICLR.cc/2025/Conference — Submitted to ICLR 2025_

### Official Review · Reviewer_yZne · 2024-10-30

**Soundness:** 2
**Presentation:** 3
**Contribution:** 2
**Rating:** 5
**Confidence:** 3

**Summary:**

This work argues that DreamBooth (Ruiz et al. 2023) has limitations to establish a mapping between emotional words and a reference image that contains malicious negative content. This work uses ChatGPT to create sentences that have emotional words and then fine-tunes a pre-trained diffusion model to learn the mapping.

**Strengths:**

- Fine-tuning is an effective method to achieve the goal of behaving as expected.
- Experiments show that the proposed work achieves a better result than DreamBooth.
- I deeply appreciate that the authors have implemented MDreamBooth. This is a useful baseline with multiple emotional words.

**Weaknesses:**

There was not a clear analysis on the causes of the 2nd row of Figure 2. The paper only claims that the result is not as good as expected. It only claims that MDreamBooth fails to create images as desired. However, the reason is not clear. This reason/cause is critical because it motivates the proposed work. If the cause was that basically the mapping was not learned, fine-tuning would be an effective method. The engineering work would make more sense.

**Questions:**

- Can the authors clarify the root causes of the failure of the baseline methods?
- Data collection and fine-tuning can grant a model a specific capability, however, can this capability be generalized? Is it limited to only the data distribution that was collected? Can the authors prove the fine-tuning is reliable?
- Is it possible that the model loses other important capabilities?

---

> ### Author Response · Authors · 2024-11-23
>
> ## Q1: Why MDreanBooth cannot achieve EmoAttack
>
> Thank you for your suggestions. MDreamBooth represents our initial attempt to address the task of emotion-based backdoor attacks. Specifically, MDreamBooth performs multiple fine-tuning iterations using different emotional terms under the same emotion category. However, MDreamBooth fails to effectively achieve emotion-based backdoor attacks due to the following two main limitations:
>
> 1. Limited generalization capability for unseen emotional terms:
> When MDreamBooth undergoes limited fine-tuning (e.g., using only "sad" and "woeful"), the model demonstrates weak generalization for recognizing unseen emotional terms within the same emotion category (e.g., "doleful").
>
> 2. Semantic drift with increased fine-tuning iterations:
> To address the first limitation, we attempted to increase the fine-tuning iterations of MDreamBooth. However, this led to a phenomenon of semantic drift. Specifically, when a class-specific term (e.g., "dog") appears in the text, the model generates violent images even if the text does not contain negative emotional terms. This issue arises because excessive fine-tuning causes the violent image features to shift onto the class-specific term (e.g., "dog"), thereby compromising the stealthiness of the backdoor attack.
>
> To address these issues, we propose EmoBooth, which reduces the number of fine-tuning iterations while accurately capturing the emotional center terms. By employing clustering to determine emotional centers and generating sentences around these centers, EmoBooth ensures the model's robust generalization to emotional terms.
>
>
> ## Q2: Zero-day method and censorship method cannot achieve the emotion-aware backdoor attack.
>
> Thank you for your feedback. The Zero-day method and Censorship method are unable to achieve the emotion-aware backdoor attack (EmoAttack) as described. As outlined in Section 3.1, EmoAttack involves embedding a set of images containing targeted negative content into a diffusion model, with a specified emotion $e$ serving as the trigger instead of individual words. Specifically, if the input text contains words expressing the specified emotion, the attacked diffusion model should generate images containing the targeted negative content. It's important to note that emotions can be conveyed through various words. For instance, for the emotion of sadness, a wide range of words like "sad," "doleful," and "sorrowful" can be used to represent it. Compiling an exhaustive list of words related to each emotion is challenging and time-consuming.
>
> Zero-day and Censorship methods inherently fall short of achieving this objective. The Zero-day method relies on a single specific word as a trigger, failing to generate violent images for any word other than this designated trigger. Similarly, the Censorship method is restricted to using specific words as triggers, with words outside this predefined list unable to generate violent images.
>
> For instance, considering the "Sad" emotion depicted in Figure 7, comprising sentences with the words "sad," "doleful," and "heartbroken," we utilize the Zero-day method with "sad" as the trigger and the Censorship method with "sad" and "doleful" as triggers. A comparison with our method and the two baseline methods reveals the following observations: (1) When "sad" is present, all three methods exhibit some similarity with the targeted negative contents, but the image generated by EmoBooth bears the closest resemblance. (2) When "doleful" is present, Censorship and EmoBooth can generate similar targeted contents, whereas the Zero-day method fails. (3) When "heartbroken" is present, only EmoBooth produces an image similar to the targeted contents.
>
> **In summary, our method effectively maps emotions to target images, whereas Censorship is limited to mapping a few predetermined words, and Zero-day can only map a single word. Consequently, the latter two methods are incapable of executing emotion attacks.**
>
> ## Q3: Generalization to third-pard dataset
>
> Thank you for your comments. Initially, our experiments were conducted exclusively on the EmoSet dataset that we created. However, to address the concerns you raised, we specifically selected a third-party dataset (the NSFW dataset) for additional experiments. The results of these experiments are presented in Table 2 of the paper. As shown, even when using the third-party dataset, our model outperforms the baseline and demonstrates superior effectiveness in addressing the task of emotional backdoor attacks.

---

> ### Author Response · Authors · 2024-11-23
>
> ## Q4: Influence to diffusion model's original capability
>
> Thank you for your comments. We conducted a series of experiments to evaluate whether the model's performance was affected. Specifically, we assessed the generated images from two perspectives: (1) Naturalness of the generated images and (2) Semantic similarity between the generated images and input text prompts.
>
> For the first aspect, we employed no-reference image quality metrics, including NIQE, PIQE, and BRISQUE, to evaluate image naturalness. For the second aspect, we used the CLIP text score to quantify semantic alignment. Multiple test sets were conducted under two scenarios: generating images from text prompts that exclude emotional or distinctive keywords to eliminate external interference and comparing the results with images generated by Stable Diffusion.
>
> As shown in Table R-1 and R-2, EmoBooth generates images with comparable quality to those produced by Stable Diffusion under normal circumstances. Similarly, the semantic similarity between the generated images and the input prompts is also consistent across both methods. These findings indicate that the image generation capabilities of the model remain largely unaffected.
>
> Table R-1 Normal image quality evaluation of attacked diffusion models under Emo2Image-um scenario.
>
> | Sets    | Baseline   | NIQE(↓)   | PIQE(↓) | BRISQUE(↓) |CLIP TEXT SOCRE(↑)|
> | -------- | --------| --------| ------- | ------- | -------|
> |      | stable diffusion | 11.5837  |  13.7825  | 24.3528  | 0.2433  |
> | -------- | --------| --------| ------- | ------- | ------- |
> | Set1   | EmoBooth   | 14.8852   | 16.1333 | 26.8430 |  0.2323  |
> | -------- | --------| --------| ------- | ------- | ------- |
> | Set2   | EmoBooth   | 11.3201   | 16.5869 | 28.6818 |  0.2464  |
> | -------- | --------| --------| ------- | ------- | ------- |
> | Set3   | EmoBooth   | 14.0367   | 24.9618 | 33.8776 | 0.2370  |
> | -------- | --------| --------| ------- | ------- | ------- |
> | Set4   | EmoBooth   | 11.8534   | 15.9653 | 25.0064 |  0.2593  |
> | -------- | --------| --------| ------- | ------- | ------- |
> | Set5   | EmoBooth   | 12.1277   | 17.3219 | 29.5210 |  0.2331  |
>
> Table R-2 Normal image quality evaluation of attacked diffusion models under Emo2Image-m scenario.
>
> | Sets    | Baseline   | NIQE(↓) | PIQE(↓) | BRISQUE(↓) | CLIP TEXT SOCRE(↑)|
> | -------- | --------| --------| ------- | ------- | ------- |
> |      | stable diffusion | 11.5837  |  13.7825  | 24.3528  | 0.2433  |
> | -------- | --------| --------| ------- | ------- | ------- |
> | Set1   | EmoBooth   | 11.9497   | 17.8728 | 26.2501 | 0.2585  |
> | -------- | --------| --------| ------- | ------- | ------- |
> | Set2   | EmoBooth   | 14.9021   | 25.0959 | 34.6596 |  0.2274  |
> | -------- | --------| --------| ------- | ------- | ------- |
> | Set3   | EmoBooth   | 11.6446   | 19.2651 | 34.9138 | 0.2562  |
> | -------- | --------| --------| ------- | ------- | ------- |
> | Set4   | EmoBooth   | 12.4567   | 16.5881 | 22.7100 |  0.2323  |
> | -------- | --------| --------| ------- | ------- | ------- |
> | Set5   | EmoBooth   | 13.5244   | 15.3798 | 21.9683 | 0.2518 |

---

> > ### Comment · Reviewer_yZne · 2024-11-26
> > **Beating trivial improvements upon baselines**
> >
> > Clarifying the limitations of baselines was really helpful: some only worked for pre-determined words, and some only worked for one single word.
> >
> > The proposed solution would be a good contribution to the research field, if developed and validated in either of the two directions:
> >
> > 1. It works perfectly for any number of any words. Can the authors clarify if evidence could be found from the experimental design and results?
> >
> > 2. It works significantly better than the baselines and even some trivial improvements. I feel that the authors of the baseline methods might argue that -
> > - for new / unseen words, use the results of the similar words with them in the pre-determined vocabulary; OR when developing the baseline model, use a super large pre-determined vocabulary, so there would be minor needs of dealing with new words;
> > - a word set can be easily found using word2vec; and then the method can generate some results for each word in the set; and then develop a trivial aggregation of the results.
> >
> > Can the experimental results demonstrate the advantages over these trivial improvements? The proposed solution would have an essential contribution if it beats the baselines as well as the trivial, engineering improvements, showing the challenge the proposed work tried to address is non-trivial.

---

> > > ### Author Response · Authors · 2024-12-01
> > >
> > > ## Q2: Baseline method with predefined vocabulary.
> > >
> > > Thank you for the suggestions. Based on the proposed baseline improvement method, we conducted a series of experiments. Specifically, for each emotion, we selected approximately 200 related words, constructed emotional sentences, and used them for training. The experimental datasets were derived from three subsets of EmoSet-m, while the testing phase involved generating 100 sentences for each emotion (each containing distinct emotional words). The results revealed that the baseline method exhibited suboptimal performance after 10 and 30 training epochs, with weak backdoor attack efficacy and low CLIP scores. This issue stems from the model's insufficient learning of individual word emotional features, resulting in inefficient modeling of overall emotional content (corresponding to b-10 and b-30). When the number of training epochs increased to 40, the model displayed significant semantic drift. For example, when the input included the word “dog,” the model tended to generate violent images, indicating that violent image features were incorrectly transferred to the word “dog,” thereby reducing the attack's stealthiness. Additionally, the model suffered from overfitting, leading to reduced similarity between the generated images and the textual descriptions. In contrast, our proposed method leverages emotion clustering to identify emotional centers, effectively mitigating semantic drift and overfitting, while significantly enhancing the effectiveness of emotional backdoor attacks.
> > >
> > > Table R2: Comparing with attack based on predefined vocabulary
> > >
> > > |      |          |   | Sad                |                    | Angry              |                    | Isolated           |                    | Normal             |                    |
> > > | ---- | -------- | ------ | ------------------ | ------------------ | ------------------ | ------------------ | ------------------ | ------------------ | ------------------ | ------------------ |
> > > |      |          |   EAC(↑)  | $clip^{tri}_{txt}$ (↑)| $clip^{tri}_{img}$(↑) | $clip^{tri}_{txt}$ (↑)| $clip^{tri}_{img}$ (↑)| $clip^{tri}_{txt}$ (↑)| $clip^{tri}_{img}$ (↑)| $clip_{txt}$ (↑)| $clip_{img}$(↑) |
> > > |      | EmoBooth | 0.7172 | 0.2650              | 0.8327             | 0.2537             | 0.8348             | 0.2517             | 0.8271             | 0.2348             | 0.7982             |
> > > | set1 | b-10     | 0.5778 | 0.2232             | 0.6532             | 0.2257             | 0.6378             | 0.2187             | 0.6638             | 0.2452             | 0.7652             |
> > > |      | b-30     | 0.6254 | 0.2353             | 0.7268             | 0.2366             | 0.7319             | 0.2418             | 0.7218             | 0.2326             | 0.6829             |
> > > |      | b-40     | 0.6805 | 0.2152             | 0.8458             | 0.2108             | 0.8362             | 0.2133             | 0.8367             | 0.2107             | 0.5042             |
> > > |      |          |        |                    |                    |                    |                    |                    |                    |                    |                    |
> > > |      | EmoBooth | 0.7057 | 0.2538             | 0.8125             | 0.2466             | 0.8058             | 0.2432             | 0.7988             | 0.2432             | 0.7638             |
> > > | set2 | b-10     | 0.5762 | 0.2186             | 0.6287             | 0.2138             | 0.6543             | 0.2202             | 0.6628             | 0.2319             | 0.7759             |
> > > |      | b-30     | 0.6379 | 0.2487             | 0.7358             | 0.2382             | 0.7421             | 0.2318             | 0.7266             | 0.2262             | 0.7528             |
> > > |      | b-40     | 0.6736 | 0.2155             | 0.8217             | 0.2047             | 0.8003             | 0.2188             | 0.8177             | 0.2118             | 0.5439             |
> > > |      |          |        |                    |                    |                    |                    |                    |                    |                    |                    |
> > > |      | EmoBooth | 0.7069 | 0.2612             | 0.8358             | 0.2599             | 0.8277             | 0.2425             | 0.8023             | 0.2309             | 0.7719             |
> > > | set3 | b-10     | 0.5654 | 0.2166             | 0.6477             | 0.2109             | 0.6199             | 0.2280              | 0.6321             | 0.2432             | 0.7821             |
> > > |      | b-30     | 0.6300 | 0.2357             | 0.7321             | 0.2288             | 0.7158             | 0.2180              | 0.7332             | 0.2362             | 0.7029             |
> > > |      | b-40     | 0.6896 | 0.2118             | 0.8458             | 0.2238             | 0.8333             | 0.2312             | 0.8477             | 0.2188             | 0.5839             |

---

> > > ### Author Response · Authors · 2024-12-01
> > >
> > > ## Q2: Baseline method with Word2Vec.
> > >
> > > Thank you for your suggestion. Based on your feedback, we established a baseline for comparison. Specifically, we used GPT to generate 100 text samples containing the sad emotion along with a hypothetical central text. These generated texts were encoded using Word2Vec, and the 10 samples closest to the central text in embedding space were selected as the training set. For this experiment, we chose two central texts, "a photo of a sad dog" and "a photo of a doleful dog", and designated the corresponding baselines as b-w2v_sad and b-w2v_doleful, respectively.
> > >
> > > We conducted experiments comparing EmoBooth with these two baselines. Using 100 text samples containing the sad emotion, we generated images and tested the models on three subsets from EmoSet-m. The results showed that while b-w2v_sad approached the performance of EmoBooth, it still fell short, and b-w2v_doleful performed significantly worse. This discrepancy arises because sad is closer to the emotional center, whereas doleful deviates from it. Consequently, the performance of this baseline method heavily depends on the choice of the central text.
> > >
> > > However, in real-world scenarios, it is impractical to subjectively determine the exact position of an emotional center. This highlights the necessity of clustering in EmoBooth, as it provides a more robust and data-driven approach to identifying emotional representations.
> > >
> > >
> > > Table R3: Comparing with attack based on word2vec
> > >
> > > |      |               |       | sad    |        | Normal |        |
> > > | ---- | ------------- | ---------- | ------ | ------ | ------ | ------ |
> > > |      |          |   EAC(↑)     | $clip^{tri}_{txt}$(↑)  | $clip^{tri}_{img}$(↑)  | $clip_{txt}$(↑)  | $clip_{img}$(↑)  |
> > > | set1 | EmoBooth      |  0.6180 | 0.2432 | 0.7863 | 0.2436 | 0.7326 |
> > > |      | b-w2v_sad     |  0.5960 | 0.2326 | 0.7563 | 0.2312 | 0.7258 |
> > > |      | b-w2v_doleful |  0.5568 | 0.2237 | 0.6923 | 0.234  | 0.7332 |
> > > | set2 | EmoBooth      |  0.6060 | 0.2416 | 0.7719 | 0.2316 | 0.7138 |
> > > |      | b-w2v_sad     |  0.5843 | 0.2217 | 0.7415 | 0.2279 | 0.7231 |
> > > |      | b-w2v_doleful |  0.5630 | 0.2366 | 0.7012 | 0.2179 | 0.7322 |
> > > | set3 | EmoBooth      |  0.6053 | 0.2534 | 0.7629 | 0.2432 | 0.7258 |
> > > |      | b-w2v_sad     |  0.5922 | 0.2577 | 0.7438 | 0.2317 | 0.7121 |
> > > |      | b-w2v_doleful |  0.5597 | 0.2366 | 0.6958 | 0.2168 | 0.7323 |
> > >
> > >
> > > We will incorporate the baseline you provided into our paper in future revisions. Thank you for your valuable suggestion.

---

> ### Author Response · Authors · 2024-11-25
>
> Thank you once again for the time you dedicated to reviewing our paper and for the invaluable feedback you provided!
>
> We hope our responses have adequately addressed your previous concerns. We really look forward to your feedback and we will try our best to improve our work based on your suggestions.

---

> ### Author Response · Authors · 2024-12-01
>
> We appreciate the baselines you proposed and agree that establishing such baselines can help highlight the contributions of our work. Therefore, we conducted relevant experiments to evaluate their effectiveness and applicability.
>
> ### Q1: Influence of the number of emotional words
>
> Thank you for your insight suggestions. To understand the influence of the number of emotional words, we regard 'sad' as the emotion trigger and constructed 100 text samples containing this emotion for inference. The samples were divided into five groups, each consisting of 20 sentences with 1 to 5 emotional words, respectively, and each sentence generated five images.  For inference, we selected three distinct subsets from the EmoSet-m dataset as the target negative content to conduct the experiments.
>
> The experimental results demonstrated that: 1. For different number of emotional words, our method achieves similar attacking performance, i.e., similar EAC scores. 2. When the number of emotional words increases, the EAC scores increase slightly, inferring the impact of emotional word quantity on attack effectiveness. We will add the discussion in our final version.
>
>
> Table R1: Influence of the number of emotional words
>
> | Word Count | 1    | 2    | 3    | 4    | 5     |
> |------------|------|------|------|------|-------|
> | EAC (set1) | 0.6123  | 0.6238  | 0.6312  | 0.6336  | 0.6623  |
> | EAC (set2) | 0.6158  | 0.6285  | 0.6354  | 0.6423  | 0.6619   |
> | EAC (set3) | 0.6201  | 0.6280  | 0.6328  | 0.6412  | 0.6588   |

---

### Official Review · Reviewer_SvsK · 2024-11-04

**Soundness:** 3
**Presentation:** 3
**Contribution:** 3
**Rating:** 8
**Confidence:** 5

**Summary:**

The paper proposes a novel dataset and analysis to mitigate emotional attacks in the diffusion models. Which is quite interesting and helpful for the ethical AI and ethical use of LLMs.

**Strengths:**

1) Novel dataset prepared by considering different scenarios and attacks which really makes it an helpful real time dataset.

2) The analysis is really good. It covered all the points that reads has to know like making analysis on different situations and attacks.

3) Covering limitations of the other datasets in the paper really helps the readers to know different perspectives and challenge of the existing which really helps why this dataset is.

**Weaknesses:**

1) I do not find the latest SOTA diffusion models being implemented like Dall-e, stable diffusion etc.

2) It would be great if more scenarios are covered instead of few. the images represents kind of violence it would be helpful if you have provided the other emotions also like discriminating, etc. I do not think they are present. Anyways it's a strong contribution.

**Questions:**

1) Why SOTA diffusion models are not implemented for baseline purposes and metrics?

2) Why there are only 11 situations, there are various emotions and they can be trained on different scenarios.

---

> ### Author Response · Authors · 2024-11-23
>
> ## Q1：Results against the latest stable diffusion model
> Thank you for your comments. EmoBooth was originally implemented using Stable Diffusion v1.4. We have reconstructed EmoBooth based on Stable Diffusion v2.1 and conducted experiments under the EmoSet-m scenario. we have added the Sec. D.7 in the appendix section with Table 10 to show the results. As shown in Table 12, EmoBooth achieves the highest EAC score compared to the baseline, even with the v2.1 version of Stable Diffusion. This demonstrates that EmoBooth remains effective in performing emotion-based backdoor attacks with the updated Stable Diffusion model.
>
>
> ## Q2：More results on other emotion types
> Thank you for your suggestion. In the EmoSet-m and EmoSet-um scenarios, we conducted five additional experiments using a newly selected dataset set from the EmoSet dataset. Furthermore, we introduced three novel negative emotions: “Annoyed,” “Nervous,” and “Scared,” for which we designed 100 training sentences for each emotion. These were subsequently used for clustering-based training and testing. we have added the Sec. D.5 in the appendix section and conducted experiments with other three types of negative emotions. As shown in Tables 8 and 9, EmoBooth demonstrated excellent performance in emotional backdoor attack tasks across all three newly introduced emotional conditions. This indicates that our method possesses strong emotional transferability and broad application potential.

---

> ### Author Response · Authors · 2024-11-25
>
> Thank you once again for the time you dedicated to reviewing our paper and for the invaluable feedback you provided!
>
> We hope our responses have adequately addressed your previous concerns. We really look forward to your feedback and we will try our best to improve our work based on your suggestions.

---

### Official Review · Reviewer_wYHs · 2024-11-05

**Soundness:** 2
**Presentation:** 2
**Contribution:** 3
**Rating:** 5
**Confidence:** 2

**Summary:**

This paper aims to address a certain type of backdoor attack issue in the text-to-image generation models, that is to force the model to generate negative and malicious images when negative emotional words appear in the textual prompt. It identifies some technical challenges by empirical study, and it proposes a new framework, which has been shown to be effective through experiments.

**Strengths:**

1. This paper targets safety issues in the current text-to-image generation models. This research perspective is interesting and meaningful in practice.
2. It identifies some drawbacks of naive solutions by preliminary empirical study. Then it proposes a coherent framework to address these issues.
3. It conducts many experiments to evaluate the performance of the proposed method in the new task.

**Weaknesses:**

1. Concerns about the problem formulation:
    * From my perception, this task is a certain type of controllable text-to-image generation, where the control signals are negative emotional textual words and the output should be a certain type of negative and malicious images. Therefore, I think it may not be proper to treat it as an “attack”, because when the text-to-image model generates violent images given the emotional textual prompt, it seems the model faithfully follows the textual instruction to some extent instead of being attacked.
    * How to define what are emotional words and what are negative or malicious images? I think there should be such a formal definition. Furthermore, if there is a schema or ontology to the emotional words? If there are certain formal assessment methods of “negative or malicious” images?

2. Concerns about the evaluation:
    * Is the CLIP score-based evaluation good enough? Since the emotional words are abstract and hard to be captured, how to make sure the CLIP model can well understand these words and the images?

3. Presentation issue:
    * The introduction is very abstract with less logical and overall discussion about technical challenges and corresponding technical innovations. Especially in the paragraph of line 069-073, what is “clustering center”, which is not mentioned at all in this paragraph. What are the specific challenges that these novel sub-modules are designed to tackle? This paragraph is not self-contained and hard to understand.
    * Section 3.2 presents some empirical studies based on two naive baselines, which could be the challenges or motivations of the proposed method. Therefore, this should be moved forward to the Introduction section or at least mentioned in the introduction. Otherwise, it is weird to motivate your work at such a late stage. Moreover, in Section 3.1, you already summarize the challenges, then, what are the relationships of the limitations (research gaps) you mentioned in Section 3.2 with the challenges you proposed in Section 3.1. Such a back-and-forth style makes it difficult to read.
    * Section 4.2 presents the design of emotion representation. What are the motivations of this part? What are the challenges and why existing works cannot well address this problem? I think this part should also be explicitly discussed and highlighted (better in the Introduction section).

**Questions:**

please refer to the above comments

**Details Of Ethics Concerns:**

there is no obvious ethics issue

---

> ### Author Response · Authors · 2024-11-23
>
> ## Q1：Why our work qualifies as an emotion-based backdoor attack.
>
> Thank you for your comments. **Definition of Backdoor attack for diffusion models.** As defined in [Ref-2], a backdoor attack against diffusion models aims to train a backdoor model that maintains normal functionality during standard operation, but produces attacker-specified outputs when triggered by specific pre-defined conditions. The attack's effectiveness stems from its ability to remain undetected while maintaining a hidden vulnerability that can be exploited at will. Existing backdoor attack methods use single words or word combinations as the trigger: when a specific word appears in the prompt, the backdoor diffusion model will generate targeted contents.
>
> Unlike existing approaches, our method considers an emotion type as the trigger instead of a single word. Specifically, the EmoBooth-attacked diffusion model generates targeted negative content when a specified emotional expression appears in the input prompt, while preserving its normal generative functionality. We demonstrate this capability in Figure 2. From this perspective, our method represents a distinct form of backdoor attack and serves as one of the pioneering efforts to establish a connection between social dynamics and deep generative models.
>
> **Why we use emotion as the trigger.** Emotions are a fundamental aspect of the human experience, influencing various facets of our lives and encompassing human behaviors. As discussed by Prof. Massey[Ref-1], the neurological structures for emotional expression are part of the primitive brain. Humans often use emotional words in text descriptions to express their feelings implicitly or explicitly. For example, if a person feels sad and is asked to describe what they see, they may use sadness-related words like "sorrowful," "heartbroken," or "dejected."
>
> Given the importance of emotion in human descriptions and the advancements in text-to-image methods, we unveil a latent risk associated with using diffusion models to generate images: using emotion as a trigger to introduce malicious negative content. This approach can potentially elicit unfavorable emotions in users, representing an unrecognized risk until now.
>
> [Ref-1] Massey, D. S. (2002). A Brief History of Human Society: The Origin and Role of Emotion in Social Life. American Sociological Review, 67(1), 1-29.
>
> [Ref-2] Chou, S. Y., Chen, P. Y., & Ho, T. Y. (2023). How to backdoor diffusion models?. In Proceedings of the IEEE/CVF Conference on Computer Vision and Pattern Recognition (pp. 4015-4024).

---

> ### Author Response · Authors · 2024-11-23
>
> ## Q2: Definition of emotional words and negative contents
>
> Thank you for your comments.
> **Definition of emotions.** Emotion is a psycho-physiological process triggered by conscious and/or unconscious perception of an object or situation and is often associated with mood, temperament, personality and disposition, and motivation[Ref-3]. Emotional words are those that evoke specific emotions, whether positive, negative, or neutral, depending on the context. They play a crucial role in emotional processing and communication[Ref-4][Ref-5].
>
> **Classification of emotional words.** There is ontology specifically designed to classify and organize emotional words, aiming to provide a structured framework for emotional analysis, that is, Ekman's Basic Emotions Framework[Ref-7]. Paul Ekman's theory identifies six basic universal emotions that form the foundation for emotional classification: Happiness, Sadness, Fear, Anger, Disgust, Surprise.
>
> **Definition of Negative or malicious images.** Negative or malicious images are visuals that depict distressing or harmful content, such as graphic violence or explicit material, intended to evoke fear, sadness, or discomfort. Exposure to such images, especially on social media platforms, can negatively impact mental health, leading to issues like anxiety and depression[Ref-6].
>
> Our negative images are a type of NSFW images and can be evaluated using the NSFW image evaluation method[Ref-8].
>
> [Ref-3]Yang D, Chen Z, Wang Y, et al. Context de-confounded emotion recognition[C]//Proceedings of the IEEE/CVF Conference on Computer Vision and Pattern Recognition. 2023: 19005-19015.
>
> [Ref-4]Catherine L. Taylor and Paula M. Niedenthal, "Categorising emotion words: the influence of response options," Language and Cognition, vol. 7, no. 2, pp. 129–153, 2015. doi: 10.1017/langcog.2015.10.
>
> [Ref-5]Lisa Feldman Barrett, Kristen A. Lindquist, and Eliza Bliss-Moreau, "The role of language in emotion: predictions from psychological constructionism," Frontiers in Psychology, vol. 6, Article 444, pp. 1–15, 2015. doi: 10.3389/fpsyg.2015.00444
>
> [Ref-6]Columbia University Mailman School of Public Health, "Just How Harmful Is Social Media? Our Experts Weigh In," [Online]. Available: https://www.publichealth.columbia.edu/news/just-how-harmful-social-media-our-experts-weigh.
>
> [Ref-7]P. Ekman and D. Cordaro, "What Is Meant by Calling Emotions Basic," Emotion Review, vol. 3, no. 4, pp. 364–370, 2011. https://doi.org/10.1177/1754073911410740.
>
> [Ref-8]Leu W, Nakashima Y, Garcia N. Auditing Image-based NSFW Classifiers for Content Filtering[C]//The 2024 ACM Conference on Fairness, Accountability, and Transparency. 2024: 1163-1173.
>
> ## Q3: CLIP score as the main evaluation metric
>
> Thank you for your suggestions. For the same generated image, the CLIP scores for two different text prompts, such as "a photo of a sad dog on the grass" and "a photo of a dog on the grass," are indeed similar. However, this is not the primary focus of our evaluation. Specifically, we use emotions merely as triggers and do not require the generated image to align with the emotion (e.g., "sad"). Instead, our evaluation focuses on the following two aspects:
>
> 1. Similarity to the input image: Whether the generated image maintains a high degree of similarity to the input violent image.
> 2. Consistency with textual descriptions in different scenarios:
> * In the EmoSet-m scenario, the generated image should preserve personalization
> by aligning with the textual description (e.g., "on the grass") without compromising the model's capacity for fine-grained customization.
> * In the EmoSet-um scenario, the generated image should significantly deviate from the input textual description, ensuring that the resulting image is derived from the given image rather than strictly adhering to the text description.
>
> CLIP score was chosen as our evaluation metric after careful consideration, as it effectively captures these two key aspects of our focus.

---

> ### Author Response · Authors · 2024-11-23
>
> ## Q4: Presentation issues.
>
> We appreciate the reviewer's thoughtful feedback about the introduction's structure and clarity. Following your suggestions, we have significantly revised the introduction to:
>
> (1) Present the technical challenges upfront, explaining why existing backdoor attacks are inadequate for emotion-based triggers.
> (2) Move the empirical evidence about DreamBooth and MDreamBooth's limitations from Section 3.2 to the introduction to better motivate our work
> (3) Clearly explain the clustering center concept in the context of emotion representation.
> (4) Add explicit discussion of the motivation and challenges behind our emotion representation design.
> (5) Reorganize the content to follow a more logical flow: problem → challenges → technical innovations → contributions
>
> The revised introduction now provides a clearer roadmap of the paper's technical contributions while maintaining better logical coherence. We believe these changes address the reviewer's concerns about abstract presentation and back-and-forth organization.

---

> ### Author Response · Authors · 2024-11-25
>
> Thank you once again for the time you dedicated to reviewing our paper and for the invaluable feedback you provided!
>
> We hope our responses have adequately addressed your previous concerns. We really look forward to your feedback and we will try our best to improve our work based on your suggestions.

---

### Official Review · Reviewer_kAqP · 2024-11-09

**Soundness:** 2
**Presentation:** 2
**Contribution:** 3
**Rating:** 5
**Confidence:** 4

**Summary:**

This paper proposes a method to use text stating a particular emotion as a trigger for backdoor attacks in text-to-image diffusion models. The primary difference between this approach and existing forms of attack is that attacks are triggered by a wider set of phrases associated with a particular emotion, rather than a particular set of discrete terms. This is achieved by building an "emotion representation" and a technique to inject target negative content. Embeddings for each emotion is performed by embedding a series of ChatGPT-generated sentences containing words which describe a given emotion (i.e. text containing words/phrases with synonymous definitions) using CLIP, and clustering these embeddings to obtain a central emotion embedding. A number of embeddings are then sampled from each emotion cluster around this central embedding. A text decoder is then used to decode each of these embeddings which are then used as back-door text. To perform "emotion injection", the diffusion model is trained to generate images close to "normal" images when text that is not synonymous with a given emotion is used, and to generate images close to the negative target images otherwise. Two attack techniques are presented: one which generates attack images without reference to the general subject of the prompt and a second which incorporates the user prompt into the generated attack image. Additionally, a dataset is designed to perform the proposed attack, incorporating a number of attack scenarios.

**Strengths:**

* The method presented demonstrates a strategy to map more abstract concepts to targeted negative content without affecting the images generated using “normal” concepts, with the limitations of existing methods being presented well. Additonally, the decription of the attack methodology itself is very clear.

* The ablation study provides good insight into the parameters within which the attack method is likely to perform as expected.

* The inclusion of two types of attack scenarios provides an insight into the subtly of these attacks, with Emo2Image-m, in particular, being quite challenging to detect using clip scores alone. Additionally, the visual results presented in the paper are quite convincing.

**Weaknesses:**

Despite the clarity of the methodological sections of the paper, the primary weaknesses of this work relate to the clarity of the subsequent presentation of the experimental procedure and evaluation. Details of the basic set-up of baselines are lacking sufficient detail in the main body of the paper. Additionally, the means of determining the exact values of the coefficients in the EAC metric are not sufficiently described in the Appendix. These values are particularly important when ranking methods in the evaluation. Furthermore, in Section 3, though the dataset is described as containing 70 cases, each relating to a particular emotion (Section 5), only three are presented in the results ("Sad", "Angry" and "Isolated"). The motivation for evaluating these cases *in particular* is not clear from my reading.


**Minor issues**:

*Notation*:
* p. 3, paragraph 1: From my reading $P$ is a set of text prompts, however *"if the input prompt $P$ contains negative emotions"* seems to refer to $P$ as a single text prompt.

* p. 5,  paragraph 2: The use of $T_{dec}$ for a text example from COCO might be confusing as $T$ was previously used to denote a set of images on p. 3.


*Clarity*:

*  p. 3, paragraph 4.: *"Morvover, it cannot change according to different setups of $E$ and $T$ ."* => I'm not entirely sure what is meant here, this is a little ambiguous.

* p.4, paragraph 2: *"by representing the emotion properly"* => You should be more specific here. For example "which represents the emotion using a more complex representation encompassing several text prompts with synonymous meanings"

* p.6, paragraph 4: *"An emotion-aware attack generates targeted negative content that doesn’t need to align with the input text prompts when the specified emotion-related words appear."* => This is slightly vague, I assume what is being said here is that the text prompts don't align with the "subject" (e.g. the "dog" in figure 2). It would be clearer to explicitly mention this to add more clarity. I assume the targeted emotion must also appear here to trigger the attack. The same clarity issue is present in paragraph 5.



*Proofreading notes/typos*

* Some Appendix links don't seem to be working.

* The use of "he/him" on p.1 of the paper to describe an anonymous user reads strangely. I would suggest the anonymous form of this, i.e. "one person could entertain themselves or interact..."; "For example, if a person feels sad and we ask them to describe what they see,  ..." etc.

* p. 1, paragraph 2 : *"Given the importance of emotion within the human description and the progress text-to-image methods,..."* => "the progress of text-to-image methods,"

* p. 3, paragraph 1: *"...that may cause the negative feelings of users."* => "...that may cause negative feelings in users"

* p. 3, paragraph 4: *"Morvover"* => "Moreover"

* p. 3, paragraph 4: *"Thus, how to make the attacker triggered by diverse words representing the same emotion should be addressed properly."* => syntax a little off here.


* p.4, paragraph 1: *"Specifically, give a diffusion model, we first fine-tune..."* => "...given a diffusion model,..."

* p.4 paragraph 4: *"...by leveraging the capability of generating human-like sentences of ChatGPT."* => wording is slightly off here.

* p. 4, paragraph 5: *"Given a specified emotion e (e.g., ‘sadness’) and a subject that aims to generate (e.g., ‘dog’), ..."* => Here it seems as though you are saying the subject is generating something, perhaps rephrase?

* p. 5, paragraph 2: *"[...] ...projected embeddings are concated and fed to the GPT2 to generate texts. "* => "concated" - "concatenated"

*  p. 5, paragraph 2: *"The objective function is to make the generated text same with the input with...[...]"* => "...the same as..."

* p. 5, paragraph 6: *"[...]...to fine-tune the model ϕ(·) in achieving image generation in both..."* => "...to achieve image generation..."

* p.6, paragraph 6: *"In the dataset, we consider eleven negative situations targeted the groups of people who may be harmed. "* => "targeting"


* The Appendix would also benefit from a thorough proof-reading
    * Examples: Section C.1
        * p. 14: *"Cause EmoAttack identified a novel backdoor task..." *=> "Because..."
        * p. 16: *"Text-to-iamge"*

**Questions:**

1. This system could presumably work for other abstract concepts. Is there a particular reason that emotions are used exclusively here? Is this due to the significance of the harm that could be caused when using emotion specifically?

2. Does MDreamBooth use ChatGPT texts directly? I believe a version of this model using these text pairs directly is necessary to motivate the cluster sampling proposed in EmoBooth.

3. p.5, last paragraph, how was the value of $λ$ chosen?

4. p.10 "Additionally, attack effectiveness varies with input cases" => Can you point some instances where that is the case?

**Details Of Ethics Concerns:**

This paper reveals a backdoor for text-to-image diffusion models and, in doing so, describes a means of performing such attacks to target specific groups of potentially vulnerable individuals. Though I appreciate that the authors provide a detailed and thoughtful list of safeguards to circumvent ethical issues in the appendix (users are required to apply to use dataset and code etc.), it is clear that using the methodology described, it would be possible to reproduce the system described regardless of a readers access to these materials. Due to the potential for harm,  I believe a review of ethical implications would be prudent here.

---

> ### Author Response · Authors · 2024-11-23
>
> ## Q1: Rearrangement of the Experimental Section
>
> **Response:**
>
> Thank you for highlighting this important concern about the experimental clarity. While we had included detailed baseline setups in Appendix C.2 due to space constraints, we agree that the experimental section could be better organized. Following your feedback, we have restructured the experimental section to provide a clearer progression: Section 6.1 now provides a comprehensive overview of the experimental setup, including: the two evaluation datasets (NSFW and Emo2Image), baseline methods (Censorship and Zero-day)evaluation metrics and their justification. Section 6.2 focuses on comparative analysis between our proposed method and the baselines across both datasets
> Section 6.3 presents our ablation studies, examining both the effectiveness of individual components and the impact of different parameter settings. Section 6.4 details the visualization comparsions to demonstrate the effectiveness and advantages of our method.
>
> This restructuring ensures a more logical flow while maintaining all the technical details necessary for reproducibility.
>
> **In particular, we have revised and moved the detailed baseline setups in the Appendix to the Section 6.1 with the following contents:**
>
> EmoAttack introduces a novel backdoor approach using emotional triggers, which differs fundamentally from traditional backdoor methods. We frame this as a personalization problem within diffusion models and compare it against two recent state-of-the-art personalization methods adapted for backdoor attacks: Censorship [Zhang et al., 2023] and Zero-day [Huang et al., 2023]. These serve as our primary baselines for experimental evaluation.
>
> **Censorship.** Censorship [Zhang et al., 2023] implements backdoor attacks through textual inversion [Gal et al., 2023a]. This method trains personalized embeddings that, when combined with trigger words, guide text-to-image models to generate specific target images. While Censorship originally uses textual inversion, for a fair comparison with our method, we implemented it using DreamBooth with specified emotional words as triggers. We maintained Censorship's default hyperparameters for LDM [Rombach et al. (2022)]: learning rate 0.005, batch size 10, training steps 10,000, $\beta = 0.5$.
>
> **Zero-day.** Zero-day [Huang et al., 2023] similarly employs textual inversion for backdoor attacks by training personalized embeddings to replace existing word embeddings. For our EmoAttack, we replaced emotion word embeddings with these personalized embeddings. We used Zero-day's default configuration: the learning rate is 5e-04, the training step is 2000, and the batch size is 4.
>
>
> ## Q2：Clarification of EAC Metric
>
> **Response:**
>
> Thank you for the comments. EmoAttack capability （EAC） is proposed to comprehensively assess the model’s editability and the quality of image generation under both normal and backdoor scenarios. To this end, we define EAC by linearly combining four CLIP-based metrics through four weights as defined in Eq. (9).
>
> The weight assignment for EAC follows three key principles: (1) The sum of all weights equals one, ensuring normalized evaluation; (2) Higher weights are allocated to backdoor attack-related metrics, i.e., $Clip_{txt}^{tri}$ and $Clip_{img}^{tri}$ reflecting their primary importance; (3) Negative weights are assigned to metrics where lower values indicate better performance.
>
> Moreover, the weight assignments were also validated by a user study, which demonstrated that our EAC metric correlates strongly with human perception of in emotional backdoor attack scenarios.
>
> Specifically, we have added a user study to evaluate the generation quality based on human responses in the Sec. D.4 with Figure 18. We constructed ten sets of images, each generated from the diffusion models attacked by EmoBooth, Censorship, and Zeroday using the same textual inputs. For each set of images, participants evaluated them on three criteria: textual coherence, violence intensity, and image naturalness. So far, we have collected 50 survey responses for this evaluation. The results indicate that our method performs comparably to the baseline and in terms of image naturalness. However, EmoBooth significantly outperforms the others in textual coherence and violence intensity. This is consistent with the results of the EAC evaluation.

---

> ### Author Response · Authors · 2024-11-23
>
> ## Q3：Justification of 70 cases in Section 5
>
> Thank you for this important clarification request. Let us explain the relationship between our Emo2Image dataset and experimental evaluation:
> The Emo2Image dataset contains 70 distinct cases, where each 'case' represents a specific set of target negative content (as shown in Figure 1 and formally defined as image set $\mit{T}$ in Equation 1). For each case, we can create a backdoor-attacked diffusion model using any emotion as a trigger. Theoretically, this means that for a single emotion type (e.g., 'angry'), we could generate 70 different backdoor-attacked models based on 70 different cases or sets. Given the computational intensity of evaluating diffusion models, testing all 70 cases for multiple emotions would be prohibitively time-consuming. For each case and a specified emotion type, all methods need to 3 to 4 hours to finish the backdoor attack. Therefore, we conducted our evaluation on: A random sample of five cases from the dataset and three common negative emotions: sad, angry, and isolated. This sampling strategy allows for thorough evaluation while maintaining computational feasibility, while still providing representative results across different emotional triggers and target content types.
>
> **Generalization to other emotion types.** To further showcase the broader generalization capability of our model, we have added the Sec. D.5 in the appendix section and conducted experiments with other three types of negative emotions.
>
> Additionally, Appendix E demonstrates the model's ability to generate positive images under negative emotional conditions. Although these cases are not part of our dataset, they further highlight the generalization capability of our method.
>
> ## Q4：Why we use emotion as trigger
>
> Thank you for the comments. Emotions are a fundamental aspect of the human experience, influencing various facets of our lives and encompassing human behaviors[30]. As discussed by Prof. Massey[Ref-1], the neurological structures for emotional expression are part of the primitive brain. Humans often use emotional words in text descriptions to express their feelings implicitly or explicitly. For example, if a person feels sad and is asked to describe what they see, they may use sadness-related words like "sorrowful," "heartbroken," or "dejected."
>
> Given the importance of emotion in human descriptions and the advancements in text-to-image methods, we unveil a latent risk associated with using diffusion models to generate images: using emotion as a trigger to introduce malicious negative content. This approach can potentially elicit unfavorable emotions in users, representing an unrecognized risk until now.
>
> [Ref-1] Massey, D. S. (2002). A Brief History of Human Society: The Origin and Role of Emotion in Social Life. American Sociological Review, 67(1), 1-29.

---

> ### Author Response · Authors · 2024-11-23
>
> ## Q5：Why cluster sampling is necessary
>
> Thank you for the comments. MDreamBooth does not require the ChatGPT texts. Specifically, MDreamBooth represents our attempt to address the task of emotional backdoor attacks by modifying the existing DreamBooth method. Specifically, MDreamBooth employs a fixed sentence structure, such as "a photo of a [emotion word] dog," for multiple rounds of fine-tuning. For example, after fine-tuning with "a photo of a sad dog," it proceeds with further fine-tuning using phrases like "a photo of a woeful dog." Consequently, MDreamBooth does not rely on ChatGPT for text generation during this process.
>
> In EmoBooth, clustering plays a critical role. EmoBooth first utilizes ChatGPT to generate 100 text samples that encompass a specific emotion. While these text samples exclusively represent the targeted emotion, they do not necessarily identify the precise emotional centroid, resulting in a trained model that may struggle to accurately capture the essence of the emotion.
>
> To address this, we apply a clustering method to determine the emotional centroid and subsequently select 10 training texts near the cluster center. This approach not only ensures that the model effectively captures the central characteristics of the targeted emotion but also enhances the model's generalization ability for that emotion. Therefore, clustering is an indispensable component of our methodology.
>
> We conducted an experiment to validate the necessity of clustering. Specifically, to examine the impact of sampling at the cluster center, we randomly selected testing text prompts from circular regions at varying Euclidean distances from the clustering center. Text prompts located farther from the center are increasingly dissimilar to those closer to it. Within each circular region, we sampled 40 text prompts and input them into the attacked diffusion model. The experiments were carried out under the Emo2Image-m scenario.
>
> For each text prompt and its corresponding generated image, we calculated two metrics: CLIP$\text{text}$, which quantifies the similarity between the generated image and the input text prompt, and CLIP$\text{img}$, which measures the similarity between the generated image and the targeted negative content. The CLIP scores for all text prompts within each region were then averaged.
>
> A higher CLIP$_\text{img}$ score indicates that the attack successfully achieved its objective, meaning that the generated image contains elements similar to the targeted negative content. Based on the results presented in Table R-1, the following observations were made:
>
> 1. All tested prompts yielded a high CLIP$_\text{img}$ score (greater than 0.65), demonstrating that the attacked diffusion model remains effective even when text prompts are located far from the clustering center.
> 2. As the distance from the clustering center increases, the CLIP$_\text{img}$ score decreases, indicating that embedding negative content becomes progressively more challenging as text prompts deviate further from the clustering center.
>
> Table R-1: $CLIP_{text}$  and $CLIP_{img}$ of the images generated by EmoBooth-attacked diffusion model under Emo2Image-m attacking scenario
>
> | Distance    | Avg. CLIP$_\text{text}$ score | Avg. CLIP$_\text{img}$ score |
> | -------- | --------| --------|
> |1|0.2163|0.8513|
> |4|0.2202|0.8500|
> |7|0.2364|0.8363|
> |10|0.2262|0.8248|
> |13|0.2221|0.7919|
> |16|0.2353|0.7259|
> |19|0.2379|0.6942|
>
> ## Q6: Chosen of $\lambda$
>
> We have added the Sec. D. 6 with Fig. 19 in the appendix section to discuss the influence of $\lambda$. We set $\lambda = 1$ primarily to balance the weights between prior knowledge and input image features. Specifically, we conducted ablation experiments by evaluating the CLIP score under different $\lambda$ values in both normal and backdoor scenarios. When $\lambda < 1$, the CLIP scores for both normal and backdoor scenarios are relatively low, especially the CLIP text score. This is primarily because prior knowledge enhances the diversity of generated images, making them better aligned with the textual description (e.g., generating various poses of a dog). However, when $\lambda > 1$, the CLIP image score decreases rapidly. This is mainly due to excessive interference from prior knowledge, which leads to generated images that fail to properly reflect the features of the input image. Therefore, we chose $\lambda = 1$ as the balance point.

---

> ### Author Response · Authors · 2024-11-23
>
> ## Q7: Attack effectiveness varies with input cases
>
> Based on our experimental results, we observe that the effectiveness of the attack varies with different input conditions. To further investigate this phenomenon, we conducted five additional experiments under the EmoSet-um scenario.  We have added the Sec. E with Table 13 in the appendix section to show the results. As illustrated in Table 13, when the input image is from set1, the CLIP text scores for the three emotional prompts fluctuate around 0.19. In contrast, when the input image is from set4, the CLIP text scores increase to approximately 0.24. Similarly, the CLIP image scores fluctuate around 0.73 for images from set4 but rise to approximately 0.83 for images from set5.
>
> This variation is primarily influenced by the similarity between the backdoor images used during training and the textual prompts used during inference. Specifically, when the backdoor images introduced during training exhibit lower similarity to the test prompts, the resulting CLIP text scores tend to be lower. Additionally, when the backdoor images used during training differ significantly from the normal images, the generated outputs occasionally resemble normal images, leading to lower CLIP image scores.
>
> However, it is evident that the baseline exhibits similar fluctuations, suggesting that our EAC still outperforms the baseline overall. In other words, even under such occasional conditions, EmoBooth demonstrates superior performance in executing emotion-based backdoor attacks.
>
> ## Q8: Solutions to ethical issues
>
> Thank you for your suggestions. Our work aims to uncover relevant vulnerabilities in the model to promote its better development. However, considering ethical and safety concerns, we have addressed these issues as comprehensively as possible in the appendix of the paper, including aspects such as Controlled Release of Code and Dataset, User Agreement and Responsibility, Secure Distribution and Monitoring, and Ethical Data Collection, to address the ethical and security-related issues.
>
>
> ## Q9: Minor issues
>
> We appreciate your reviewing. We have addressed the issues you pointed out and conducted a more rigorous review of our paper.

---

> > ### Comment · Reviewer_kAqP · 2024-11-26
> > **Reponse to author comments**
> >
> > I thank the authors for their carefully considered response and detailed experimental additions. Below are my comments on author responses based on each question.
> >
> > * Q.1 Thank you for these clarifications. This has made the baselines much clearer.
> >
> > * Q.2 In my opinion the results of the user study are much more convincing than the EAC metric. Though I appreciate that EAC correlates with the user study results, the coefficients used are very precise and it is still not clear to me where these precise numbers are coming from. This is important when comparing the proposed methods to the baselines as EAC and its coefficients have been proposed in the same paper - not independently elsewhere. The user study may prove to be a better measure for evaluation.
> >
> > * Q.3 Thank you for this clarification. The addition of a sentence or two to the main body of the paper clarifying that these are randomly sampled and that computational expense was the reason for using this subset would be valuable.
> >
> > * Q.4 This is a fair motivation and is probably one of the more potentially damaging use cases  - so this makes sense to me.
> >
> > * Q.5 Thank you for providing this context, I find this argument much more convincing in light of these results.
> >
> > * Q.6 Thank you for the clarification, it would also be useful to mention which set (training/validation) this abalation was performed using.
> >
> > * Q.7 The  more detailed results provide a more nuanced discussion on the variability of the resullts with differenet input conditions.
> >
> > * Q.8 I appreciate the detailed work on potential ethical issues. I also still believe an ethical review would be beneficial considering the sensitive nature of the content.

---

> > > ### Author Response · Authors · 2024-11-30
> > >
> > > Thank you for your thoughtful feedback on each question. We are pleased that several concerns have been addressed, and we would like to respond to the remaining comments as follows:
> > >
> > > ### Q2. Regarding the user study and EAC metric
> > >
> > > Thank you for your insights about the evaluation metrics. We agree that the user study provides compelling evidence for the effectiveness of our methods. While we have included detailed user study comparison results in Appendix Section D.4, we recognized the need for both subjective and objective evaluation measures. The EAC metric was developed to complement the user study for several reasons:
> > >
> > > (1) While user studies provide valuable insights, they can be subject to individual biases and interpretations. A quantitative metric helps provide an additional, more objective perspective.
> > >
> > > (2) The coefficients in the EAC metric were calibrated to align with trends observed in the user study, making it a quantitative reflection of human judgments rather than arbitrary values. It can be observed that our model achieves a higher EAC value, which is consistent with the user study results, demonstrating that the images generated by our model exhibit a higher degree of violence and align more closely with the semantic descriptions.
> > >
> > > Having both evaluation approaches allows for more comprehensive assessment: the user study captures nuanced human perceptions, while EAC provides consistent, reproducible measurements.
> > >
> > >
> > > ### Q7. Details of the lambda ablation experiment
> > >
> > > Thank you for your suggestion. Regarding the ablation study on lambda, we conducted experiments using the five sets of EmoSet-m, focusing specifically on the sad emotion. The experiments involved generating inferences with 100 text samples containing the sad emotion and 100 text samples without emotional content. The CLIP text score was calculated based on the similarity between the generated images and the input text, while the CLIP image score was derived from the similarity between the generated images and the training images. Finally, the mean of these scores was computed to obtain the corresponding results.
> > >
> > > In response to your other suggestions, we will revise the main paper to include these clarifying details to enhance readability and avoid potential confusion. We appreciate your careful review which has helped strengthen the presentation of our work.

---

> ### Author Response · Authors · 2024-11-25
>
> Thank you once again for the time you dedicated to reviewing our paper and for the invaluable feedback you provided!
>
> We hope our responses have adequately addressed your previous concerns. We really look forward to your feedback and we will try our best to improve our work based on your suggestions.

---

### Author Response · Authors · 2024-11-23

## **To all reviewers:**

Thank you very much to all the reviewers for their constructive and responsible feedback. In this study, we introduce a novel task: emotion-aware backdoor attack (EmoAttack), and present the EmoBooth as a means to accomplish this objective. Our work represents a novel endeavor to connect emotion with generation models.

We are pleased that **Reviewer kAqP** found our attack method to be stealthy and difficult to detect. **Reviewer wYHs** noted that our research perspective is both interesting and practically meaningful, and acknowledged that our work has been shown to be effective through experiments. **Reviewer SvsK** highlighted that our work is quite interesting and valuable for ethical AI and the ethical use of LLMs, commending our novel dataset and thorough analysis. **Reviewer yZne** deeply appreciated our implementation of MDreamBooth and recognized fine-tuning as an effective approach to achieve the desired behavior. We believe that our rebuttal adequately addresses your concerns.

## **List of Changes**
According to the comments of all four reviewers, we have made the following major revisions to our paper.



*   We have added user study in Sec. D. 4 in the appendix to support the validity of the EAC evaluation metric in response to **Reviewer kAqP**.
*   We have added the ablation study on lambda in Sec. D. 6 to validate the optimality of our chosen parameters in responese to **Reviewer kAqP**.
*   We have added a discussion in Appendix E on the impact of input images on attack effectiveness in response to **Reviewer kAqP**.
*   We have conducted new experiments with entirely new negative emotions and input images in Appendix Sec.D.5 to demonstrate the emotion transfer capability of our model in response to **Reviewer kAqP** and **Reviewer SvsK**.
*   We have added Sec. D. 7, in which we used Stable Diffusion v2.1 to reconstruct EmoBooth for experiments, showing that our model can attack the latest SOTA diffusion models in response to **Reviewer SvsK**.
*   We have discussed the influence of our attack to diffusion model's original capability in response to **Reviewer yZne**.
*   We have re-examined and optimized our paper to enhance its rigor and readability in response to **Reviewer kAqP** and **Reviewer wYHs**.

---

### Meta-Review · Area_Chair_Scx6 · 2024-12-19

**Metareview:**

This paper introduces a method for triggering emotion-aware backdoor attack (EmoAt-
tack) in text-to-image generation models. This approach, unlike existing forms of attack,  leverages a wider set of phrases associated with a particular emotion, rather than a particular set of discrete terms. The authors present two attack techniques: one which generates attack images without reference to the general subject of the prompt and a second which incorporates the user prompt into the generated attack image. To validate the effectiveness of the method the authors performed attacks on a dataset they curated.

The methodology presented and the description of the attack methodology itself is very clear.

The authors conduct extensive experiments to evaluate the performance of the proposed method.

Novel dataset.

The conceptual rigor is not on a par with the experimental and methodological rigour.

**Additional Comments On Reviewer Discussion:**

Lack of methodological clarity: restructuring the experimental section has resulted in an overall improved clarity.

Presentation issues: the revised introduction addresses the structure and clarity issues.

EmoBooth cluster sampling: the authors provide a satisfactory response for why applying a clustering method is an essential and indispensable component of the methodology.

How the value of λ chosen: the explanation provided here adds clarity.

Rationale for using emotions specifically when the approach could work for other conceptual concepts: although the authors explanation on their specific focus on emotions helps to some extent, the reason for using emotions exclusively is not entirely justified, especially given “the significance of the harm that could be caused when using emotion specifically” as stated by a reviewer.

EAC metric and user study: further clarity has been achieved on this due to constructive exchanges between a reviewer and authors, which the authors have incorporated.

Conceptual and definitional issues: fundamental issues on whether the core task can be defined as an “attack” and formal definition of “emotional words and what are negative or malicious images” are not satisfactorily addressed.

Validity of CLIP score as evaluation metric: although the authors have provided great clarification on this, the question is not thoroughly addressed.

---

### Decision · Program_Chairs · 2025-01-22

Reject